# Identification of a pro-angiogenic functional role for FSP1-positive fibroblast subtype in wound healing

Sarika Saraswati [ID] [1], Stephanie M.W. Marrow[1], Lester A. Watch[1] & Pampee P. Young[1,2]

Fibrosis accompanying wound healing can drive the failure of many different organs. Activated fibroblasts are the principal determinants of post-injury pathological fibrosis along with physiological repair, making them a difficult therapeutic target. Although activated fibroblasts are phenotypically heterogeneous, they are not recognized as distinct functional entities. Using mice that express GFP under the FSP1 or αSMA promoter, we characterized two non-overlapping fibroblast subtypes from mouse hearts after myocardial infarction. Here, we report the identification of FSP1-GFP+ cells as a non-pericyte, non-hematopoietic fibroblast subpopulation with a predominant pro-angiogenic role, characterized by in vitro phenotypic/cellular/ultrastructural studies and in vivo granulation tissue formation assays combined with transcriptomics and proteomics. This work identifies a fibroblast subtype that is functionally distinct from the pro-fibrotic αSMA-expressing myofibroblast subtype. Our study has the potential to shift our focus towards viewing fibroblasts as molecularly and functionally heterogeneous and provides a paradigm to approach treatment for organ fibrosis.

[1] Department of Pathology, Microbiology, and Immunology, Vanderbilt University Medical Center, Nashville, TN 37232, USA. [2] American Red Cross, National Headquarters, Washington, DC 20006, USA. Correspondence and requests for materials should be addressed to S.S. (email: sarika.saraswati@vumc.org) or to P.P.Y. (email: pampee.young@redcross.org)

Fibrosis is the typical response to injury that leads to distorted tissue architecture, pathologic signaling and ultimately organ dysfunction[1]. In response to tissue injury, a population of fibroblasts activates, proliferates, and plays a vital part in the entire repair process: from the generation of granulation tissue, revascularization and restoration of cardiac architecture, and function to tissue remodeling and eventually scar formation[2]. Reduced fibroblast function may lead to ventricular dilatation and catastrophic rupture due to failure to repair the infarcted region[2,3]. Alternatively, continued and unchecked fibroblast activity results in fibrosis, adverse cardiac remodeling, and ultimately heart failure[4].

It is increasingly recognized that fibroblasts are a phenotypically and functionally diverse population of cells that must be studied in a temporal and tissue/injury-specific context[5–7]. In an uninjured heart, for example, fibroblasts are quiescent and express surface markers such as vimentin, Tcf21, and PDGFRα[8]. Following injury, resident fibroblast populations expand and express proteins unique to their activated state[2,9]. Two markers expressed in the post-injury (activated) fibroblasts are fibroblast specific protein 1 (FSP1; S100A4) and α smooth muscle actin (αSMA); however, neither marker is exclusively specific to fibroblasts[10–13]. These markers are absent in the quiescent fibroblasts of heart, kidney, and lung[11,12,14]. αSMA protein, expressed in myofibroblasts, has long been studied as a hallmark of pathological fibrosis. Less is known about the significance of FSP1, a member of the S100 family of intracellular calcium-binding proteins. Several studies have confirmed FSP1 as a marker of injury-induced fibroblasts[12,13,15]. Interestingly, FSP1+ and αSMA+ fibroblasts have been reported to be non-overlapping fibroblast populations in the injured heart, skin, liver, and kidney[5,6,11–13,16], but their population dynamics as well as their functional and molecular differences in the context of tissue repair are lacking.

We hypothesize that functional and molecular characterization of the major post-injury fibroblast subtypes is necessary to target the pathological scarring/fibrotic effects of fibroblasts without impeding their pro-reparative functions. In this study, we used transgenic mice in which FSP1 and αSMA-expressing cell populations were genetically tagged with GFP. We isolated these fibroblast subpopulations from the injured murine heart, taking specific care to exclude non-fibroblast populations, which are also known to express the FSP1 protein. Detailed molecular and functional studies are done to identify the unique roles of FSP1 and αSMA-expressing fibroblasts in injury repair.

## Results

**FSP1 and αSMA are present in distinct cells post injury**. We evaluated the expression pattern of FSP1 and αSMA proteins in uninjured organs and at various time points following tissue injury. For cardiac injury (Fig. 1a), myocardial infarctions (MIs) were induced by ligation of the left anterior descending coronary artery (LAD) in mice[17]. Human cardiac tissue was collected postmortem from cadavers at least 2 weeks post infarct (Fig. 1d). For skin injury, excisional (1 cm) full-thickness wounds (Fig. 1b) were generated on the dorsum[18]. Kidney injury (Fig. 1c) was induced by unilateral ureteral obstruction by ligating the ureter just distal to the renal pelvis[19]. FSP1 and αSMA protein expression and localization with respect to injury were evaluated by co-immunofluorescence (IF) using confocal microscopy. Consistent with published studies, uninjured heart, skin, and kidney tissues were negative for anti-FSP1 staining[11–13]. Anti-αSMA staining in uninjured and at early time points after injury was restricted to vascular structures (such as pericytes and vascular smooth muscle cells). FSP1+ stromal cells were detectable within 48 h of injury in

heart, at day 4 in skin, and at day 7 in the injured kidney parenchyma, whereas significant αSMA expression (excluding perivascular cells) appeared between 7 and 8 days after injury in all three models. In the cardiac infarct model the abundance of cells expressing both markers declined significantly (or were completely absent) by day 30 after the generation of stable scar. No co-localization or overlap between these two cell populations was identified in the heart, kidney, or skin at any post-injury time point evaluated. FSP1 has been noted by multiple groups to also mark hematopoietic (myeloid) cells[10,16] and endothelial cells[8]. We confirmed that a population of both hematopoietic and endothelial cells expressing FSP1 were present in injured mouse left ventricles (LV) at post-MI d8 (Supplementary Fig. 1A). However, around 15% of the GFP+/FSP1+ cell population did not stain with CD45 or CD31 as identified through flow cytometry analysis and confocal microscopy, indicating the presence of a non-hematopoietic and non-endothelial FSP1-expressing cell subset (Supplementary Fig. 1, A, B).

To further study the degree to which FSP1+ cells originated from the bone marrow, FSP1-GFP chimeric mice were generated that had undergone ablative bone marrow transplantation (BMT) from syngeneic C57BL/6 mice. The FSP1-GFP reporter mice (gift from Dr. Eric Neilson) express GFP under the FSP1 promoter. However, the FSP1-GFP chimeric animals, following engraftment with C57BL/6 marrow, would cease to express GFP from any bone marrow-derived cells. These chimeric animals enabled us to evaluate the degree to which the number of GFP+ cells present in the injured site were the result of inflammatory cells migrating into the injury site from circulation. As expected, there was a significant reduction in GFP+ hematopoietic cells in the injured myocardium as identified by CD45 expression alone (from 42.9 to 2.4%) or by CD45/CD31 co-staining (from 34.8 to 7.75%) in BMT GFP-FSP1 mice in comparison to non-BMT GFP-FSP1 mice (Supplementary Fig. 2). These results indicated that a substantial number of GFP-expressing FSP1+ cells in the injured heart were indeed hematopoietic cells originated from bone marrow. After exclusion of the BM-derived FSP1+ population, we observed a relative increase in non-hematopoietic GFP+/FSP1+ cells, lacking hematopoietic markers from 14.7 to 65.8% in BMT GFP-FSP1 chimeric mice, demonstrating that a majority of the FSP1+ fibroblasts originated at the site of injury (Supplementary Fig. 2). These data confirmed that the injured heart contained substantial numbers of both FSP1+ fibroblasts and non-fibroblasts (hematopoietic and endothelial cells) post injury. In addition, most of the non-hematopoietic cells originate at the site of injury and are not recruited from the bone marrow.

**FSP1+ and αSMA+ fibroblast isolation from the injured heart**. Our data above support published reports that FSP1+ and αSMA+ cells are indeed unique post-injury fibroblast populations[5,6,11–13,16]. Little is known about the physiologic characteristics and cellular roles of these two subsets. To identify the cellular, molecular, genomic, and functional features of these fibroblast subtypes, we isolated FSP1 and αSMA-expressing cells from two distinct reporter mice that express GFP under FSP1 (FSP1-GFP) or αSMA (αSMA-GFP) promoters (gifts from Drs. Eric Neilson and Ivo Kalajzic, respectively) from left ventricle of mice post infarct[20,21]. Since we observed peak endogenous expression of FSP1 and αSMA protein between day 8 and 15 after experimental MI (Fig. 1a), we selected that time frame post MI (i.e., d10 post MI) to isolate each respective post-injury GFP+ fibroblast cell type by fluorescence assisted cell sorting (FACS) (Fig. 2a). To better understand the reporter fidelity of the αSMA mouse model, we evaluated the percentage of GFP+ cells which co-expressed αSMA and found that to be very high at both 4 and

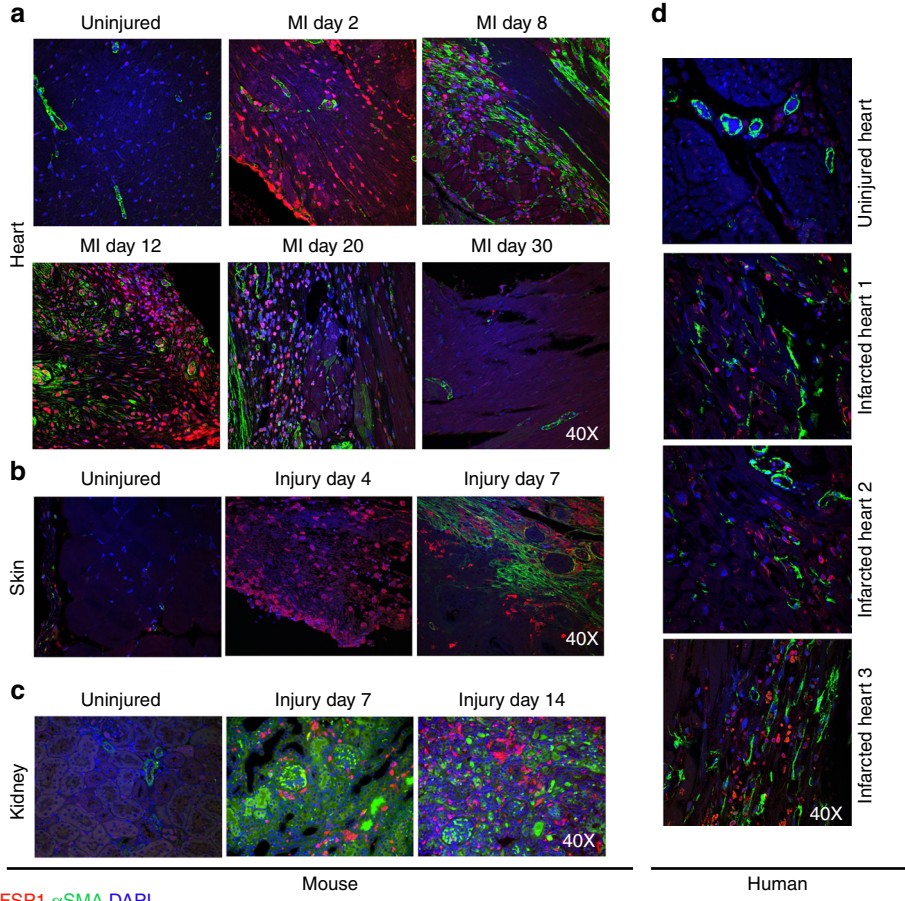

**Fig. 1** FSP1 and αSMA mark distinct populations after multiple types of tissue injury. **a** Confocal analysis of FSP1+ and αSMA+ cell populations from heart 2, 8, 12, 20, and 30 days post myocardial infarction (MI; ×40). Uninjured heart was used as control ($n = 3$). **b** Confocal analysis of representative 1 μM histologic sections from murine skin stained for FSP1+ and αSMA+ fibroblast populations 4 and 7 days after excisional full-thickness cutaneous wound. Uninjured skin was used as control (×40) ($n = 3$). **c** Confocal analysis of representative histologic sections of murine kidney stained for FSP1+ and αSMA+ fibroblast populations 7 and 14 days after unilateral ureteral ligation. Contralateral uninjured kidney was used as control (×40) ($n = 3$). **d** FSP1 and αSMA staining on histological sections of uninjured ($n = 1$) and post myocardial infarction human hearts ($n = 3$)

10 days after MI (88% ± 0.7 standard deviation (SD) and 99.9% ± 0.05 SD, respectively, $n = 3$ for uninjured, d4 and d10 post MI, Supplementary Fig. 3). Interestingly, the percentage of αSMA + cells which were GFP + was less at day 4 (19.5% ± 0.6 SD) and rose to 94% ± 7 SD at day 10 (Supplementary Fig. 3), suggesting that GFP expression lagged somewhat behind αSMA expression. Importantly, during the time frame of d10 post MI used for αSMA cell isolation, there was high concordance of αSMA cells which expressed GFP and vice versa. As expected, negligible number of cells were positive for GFP and αSMA co-staining in uninjured heart (Supplementary Fig. 3). The expression pattern of FSP1 and GFP protein in GFP-FSP1 mice has been well characterized elsewhere[10,20].

Since the uninjured hearts had no detectable GFP+ cells in both murine models, these cells isolated from uninjured mice were used to establish the background signal in the GFP channel post compensation for all experiments (Fig. 2b). A gate was drawn to capture all cells producing a larger voltage pulse area than the established background in the GFP channel. Both FSP1+ and αSMA+ cells were sorted based on GFP expression. To eliminate contaminating FSP1+ hematopoietic and endothelial cells, CD45 (GFP+; 12–16%) and CD31 (GFP+; 8–15%) cells were excluded during our sorting for FSP1+ fibroblasts (GFP+/CD45−/CD31−; 10–20%)[8] (Fig. 2c, Supplementary Figs. 5 and 6B, and Supplementary Table 4). A negligible amount of endothelial (GFP + /CD31 +

cells; 3.8% ± 0.0164 SD; $n = 5$) and hematopoietic (GFP + /CD45 + cells; 3.18% ± 0.0112 SD; $n = 5$) cells co-express GFP in injured αSMA-GFP mouse hearts and were not excluded during GFP sorting (GFP+, 15–20%) from αSMA-GFP mouse hearts (Fig. 2b, Supplementary Figs. 4 and 6A, and Supplementary Table 3). Further immunostaining of the sorted populations of both FSP1 and αSMA ensured that CD45 and CD31 cells were absent from both fibroblast isolations (Fig. 2d). Although, we took substantial measure to ensure cellular purity of FSP1 and αSMA cell population, we cannot rule out the caveats of using transgenic mouse models to isolate cell types based on transgene expression. The isolation may contain rare, contaminating cells that do not represent fibroblasts. Following sorting, our first step was to confirm the identity of GFP-expressing FSP1+ and αSMA+ cells as fibroblasts. FACS sorted cells were plated on coverslips and confirmed by IF to express FSP1 antigen and to lack expression of αSMA as well as hematopoietic and endothelial markers, CD45 and CD31 (Fig. 2d). GFP+/CD45+ or GFP+/CD31+ cell populations were used as positive controls (Supplementary Fig. 7). To support our observations from histologic immunolocalization in injured tissues, FACS sorted GFP+ cells (Fig. 2b, c) from the injured heart of αSMA-GFP and FSP1-GFP reporter mice were negative for FSP1 antigen or αSMA antigen, respectively, but exhibited robust expression of their respective proteins, αSMA, or FSP1 (Fig. 2d). Both FSP1+ and αSMA+ cells expressed Collagen type 1 alpha-1

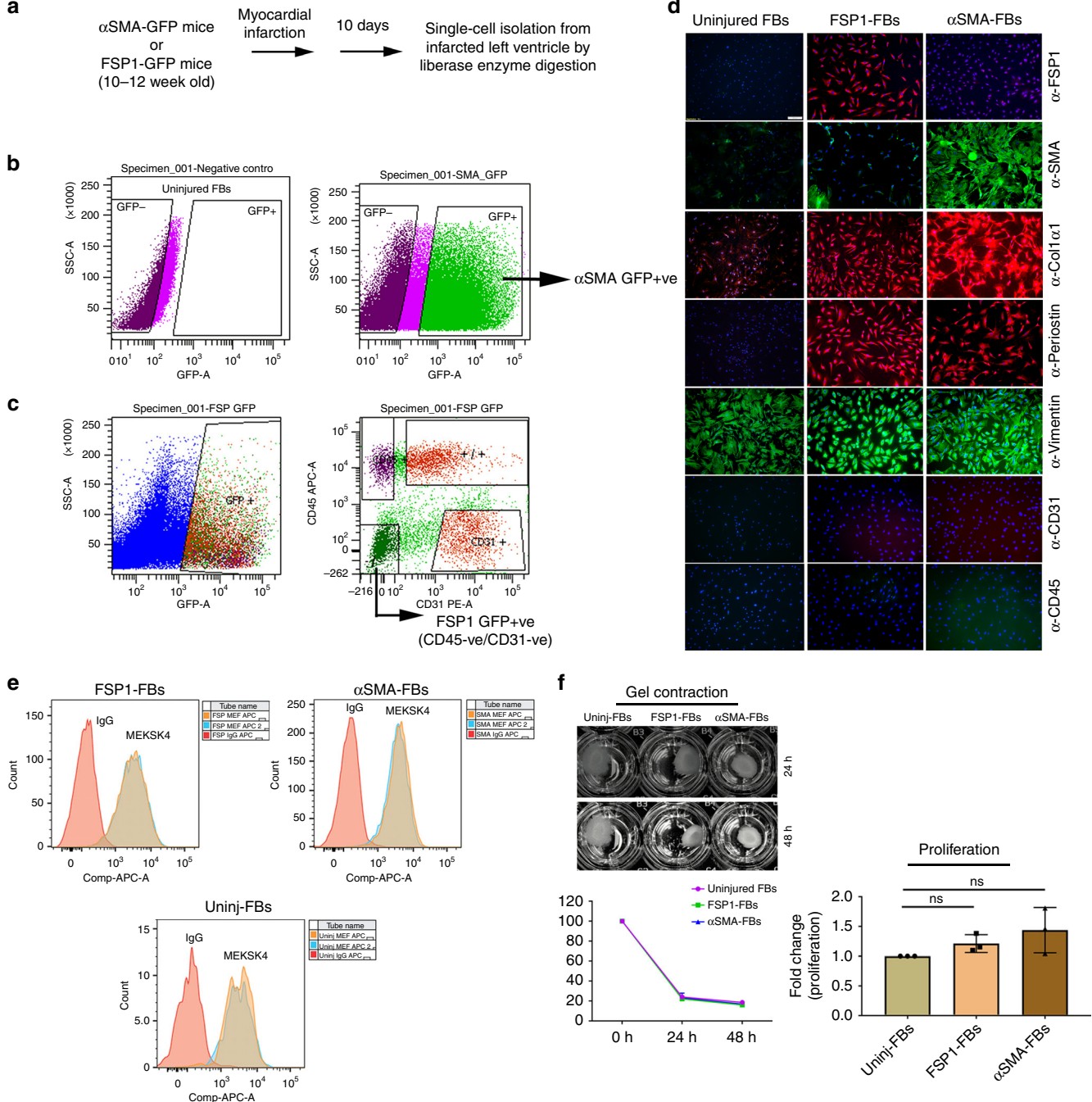

**Fig. 2** FSP1$^+$ and αSMA$^+$ cells exhibit the molecular phenotype and functionality of fibroblasts. **a** Schematic of activated fibroblast isolation 10 days post myocardial infarction (MI) from murine hearts. **b**, **c** Representative dot plot for fluorescence assisted cell sorting (FACS) of FSP1-expressing (GFP$^+$/ CD31$^-$/CD45$^-$) and αSMA-expressing (GFP$^+$) fibroblasts from FSP1-GFP and αSMA-GFP mice left ventricle 10 days after MI. Uninjured fibroblasts were used as unstained negative control. CD45 and CD31 antibodies were used to gate out hematopoietic and endothelial cells, respectively. **d** Immunofluorescence staining of GFP$^+$ populations (P0) from left ventricle of uninjured, FSP1-GFP, or αSMA-GFP sorted by FACS showed FSP1$^+$ and αSMA$^+$ as unique fibroblast populations, which express fibroblasts markers, such as COL1α1, periostin, and vimentin, but not hematopoietic marker CD45 or the endothelial marker CD31. Nuclei were stained with DAPI ($n = 3$). Scale bar = 100 μM; ×10 magnification. **e** Representative FACS overlay histogram of uninjured, FSP1$^+$, and αSMA$^+$ fibroblasts (P3–P5) to evaluate expression of MEF-SK4. Rat IgG was used as negative control ($n = 2$). **f** Representative picture of collagen gel contraction in the presence of uninjured, FSP1$^+$, and αSMA$^+$ fibroblasts (P3–P5) (left panel, top). Percentage change in the initial gel area following 24 and 48 h of contraction in the presence of uninjured, FSP1$^+$, and αSMA$^+$ fibroblasts (left panel, bottom) ($n = 2$). Fold change in proliferative ability of FSP1$^+$ and αSMA$^+$ fibroblasts (P3–P5) compared with uninjured fibroblasts measured by Brdu proliferation assay (right panel). $^{ns}p > 0.05$ was calculated by one-way ANOVA, $n = 3$ experiments were performed; ns = not significant. Cell isolation and FACS were performed at least three independent times for staining and proliferation assay and two independent times for contraction assay and MEF-SK4 flow analysis. For each sorting, cells were isolated from pooled homogenates from three to four injured murine hearts

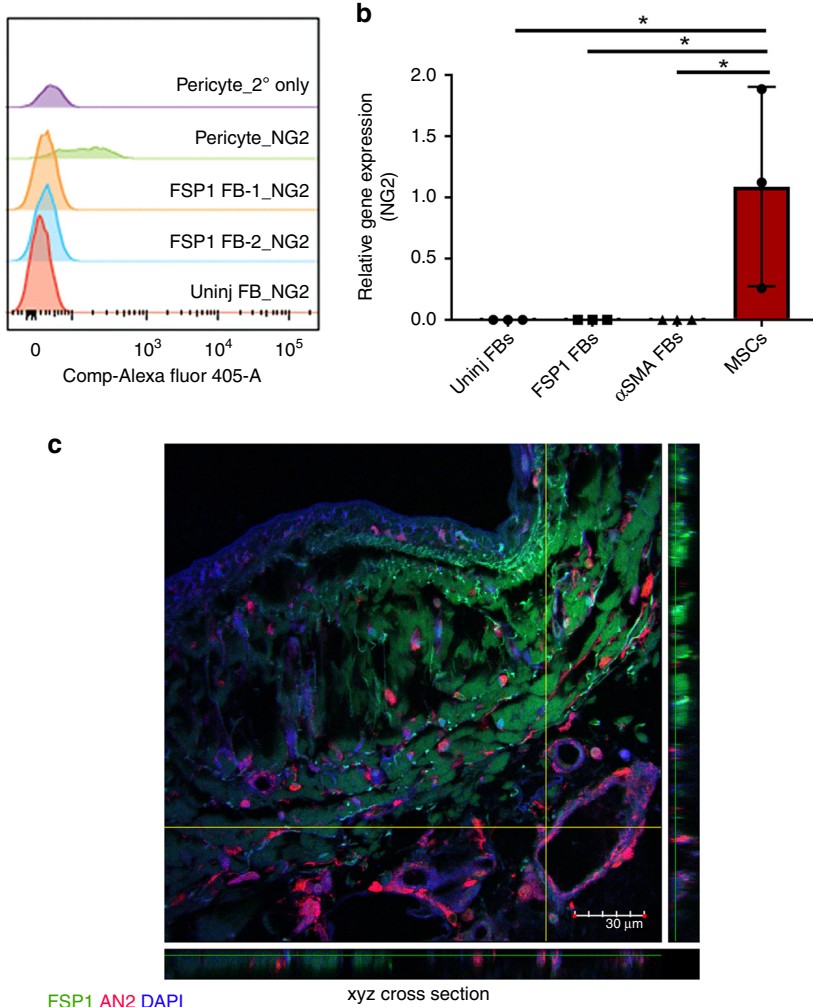

**Fig. 3** FSP1$^+$ fibroblasts do not express pericyte marker, AN2/NG2. **a** Staggered FACS histogram representing AN2/NG2 expression in FSP1$^+$ and uninjured fibroblasts (P3–P5). Human retinal pericytes were used as positive control and secondary only antibody was used as a negative control. **b** Relative expression of *An2/Ng2* in cultured uninjured, FSP1$^+$, and αSMA$^+$ fibroblasts (P3–P5). $p < 0.05$ was calculated by one-way ANOVA, $n = 3$ experiments were performed independently; bar represents mean ± SD. Mesenchymal stem cells (MSCs) were used as a positive control. **c** Confocal analysis of FSP1 (green), AN2/NG2 (red), and DAPI (blue) staining on post myocardial infarcted human heart; $n = 2$

chain (COL1α1), periostin, and vimentin by IF analysis. On the other hand, uninjured fibroblasts expressed vimentin but lacked expression of activated fibroblast markers: FSP1, αSMA, periostin, and COL1α1. Ultrastructural evaluation of fibroblast subtypes showed abundance of rough endoplasmic reticulum, golgi apparatus, and mitochondria with elongated nucleus in αSMA$^+$, FSP1$^+$, as well as uninjured fibroblast subtypes as described earlier for fibroblasts and myofibroblasts (Supplementary Fig. 8A)[22]. Primary mouse microvascular endothelial cells, used as a control, appeared quite distinct from the fibroblast subtypes with dense cytoplasm and fewer numbers of intracellular organelles. Moreover, around 95–97% of all three fibroblast subtypes stained positive for fibroblast marker MEF-SK4 by flow cytometry (Fig. 2e)[23] and expressed transcripts of fibroblast specific markers *Pdgfrα*, *Col3*, *Col1α1*, and *Fn1* as assessed by semiquantitative real-time PCR (Supplementary Fig. 8B). Also, none of the three fibroblast subtypes expressed the pericyte marker, AN2 (a mouse homolog of the rat NG2), determined by semiquantitative real-time PCR and flow analysis (Fig. 3a, b, Supplementary Figs. 4 and 5, Supplementary Tables 3 and 4). In fact, confocal analysis of injured human heart failed to indicate any co-localization between FSP1 and AN2/NG2 (Fig. 3c). Interstitial as well as perivascular location of AN2-positive cells was

evident in the injured heart, whereas FSP1$^+$ fibroblasts were primarily localized in the interstitial space. These results determined that isolated fibroblast subtypes, specifically the less characterized FSP1$^+$ subtype, are distinct from injury-activated pericytes. Importantly, FSP1$^+$ (GFP$^+$/CD45$^-$/CD31$^-$) fibroblasts exhibited similar levels of collagen gel contractility and proliferative activity (Fig. 2f) as shown by both uninjured fibroblasts and αSMA$^+$ fibroblasts. These results demonstrated successful isolation of activated FSP1$^+$ (GFP$^+$/CD45$^-$/CD31$^-$) and αSMA$^+$ fibroblasts from injured hearts using two distinct reporter mice. Their identity as fibroblasts was confirmed by molecular, ultrastructural, phenotypic, and functional characterization.

**FSP1$^+$ fibroblasts do not differentiate into αSMA fibroblasts**. Our data demonstrated no cellular co-localization between FSP1 and αSMA-expressing cells in three different tissue injury models. Since many studies have demonstrated undifferentiated fibroblasts to differentiate into αSMA-expressing myofibroblasts in defined medium, we assessed if FSP1-expressing fibroblasts could be similarly induced[24,25]. We cultured FSP1-expressing as well as uninjured fibroblasts in the presence of TGFβ in conditions

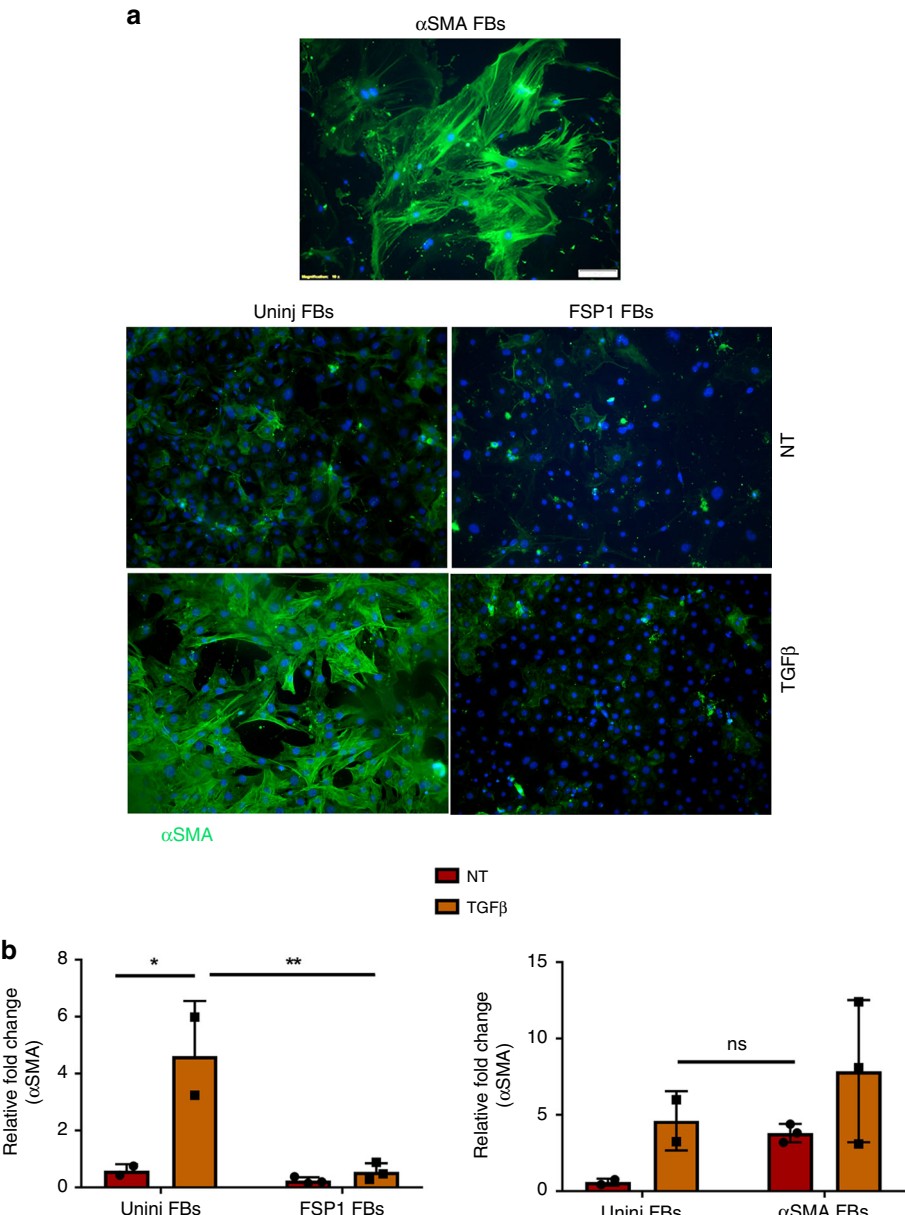

**Fig. 4** FSP1[+] fibroblasts do not differentiate into αSMA-expressing fibroblasts. **a**. Immunofluorescence staining for αSMA protein in fibroblasts (P0) isolated from uninjured and FSP1-GFP mice hearts 10 days post injury and cultured for 72 h in the presence or absence of TGFβ. Nuclei were stained with DAPI ($n = 3$). αSMA[+] fibroblasts isolated from αSMA-GFP mice post injury were used a positive control (P0). Scale bar = 100 μM; ×10 magnification **b** Relative fold change of αSMA expression in both uninjured fibroblast and FSP1[+] fibroblasts (P3–P5) cultured with or without TGFβ, *$p < 0.05$; **$p < 0.005$; ns$p > 0.05$ was calculated by two-way ANOVA, $n = 2$ experiments for uninjured fibroblasts and $n = 3$ experiments with FSP1[+] and αSMA[+] fibroblasts were performed; bar represents mean ± SD; ns = not significant

previously published as permissive for myofibroblast differentiation[24]. As expected, primary fibroblasts from uninjured hearts differentiated into αSMA-expressing myofibroblasts identified by immunofluorescence (Fig. 4a) and semiquantitative real-time PCR analysis (Fig. 4b) whereas FSP1-expressing fibroblasts failed to differentiate into myofibroblasts in vitro. αSMA-expressing fibroblasts were used as a positive control. These results further confirmed that FSP1 and αSMA-expressing fibroblasts were two distinct activated fibroblast subtypes, which did not change phenotype.

**FSP1[+] and αSMA[+] fibroblasts are transcriptionally distinct.** To identify the transcriptomic signature variability between these cell types, we employed a semiquantitative real-time PCR-based fibrosis array (Fig. 5a). Following the analysis of 84-gene panel PCR array, we identified distinct molecular signatures between αSMA[+], FSP1[+], and uninjured fibroblasts. Genes involved in matrix synthesis were upregulated in both FSP1[+] and αSMA[+] fibroblasts; although the level of these genes was much higher in αSMA[+] vs. FSP1[+] fibroblasts (Fig. 5b). Interestingly, Gremlin1 (*Grem1*) which is known to modulate angiogenesis was significantly elevated (4.75 fold) in FSP1[+] in comparison with αSMA[+] fibroblasts (Fig. 5c). The transcript levels of matrix metalloproteases 2 and 13 (*Mmp2* and *Mmp13*) and pro-angiogenic factor *Vegf* were also modestly elevated in FSP1[+] fibroblasts vs. αSMA[+] fibroblasts. In addition, an array of transcripts including pro-fibrotic cytokines were significantly

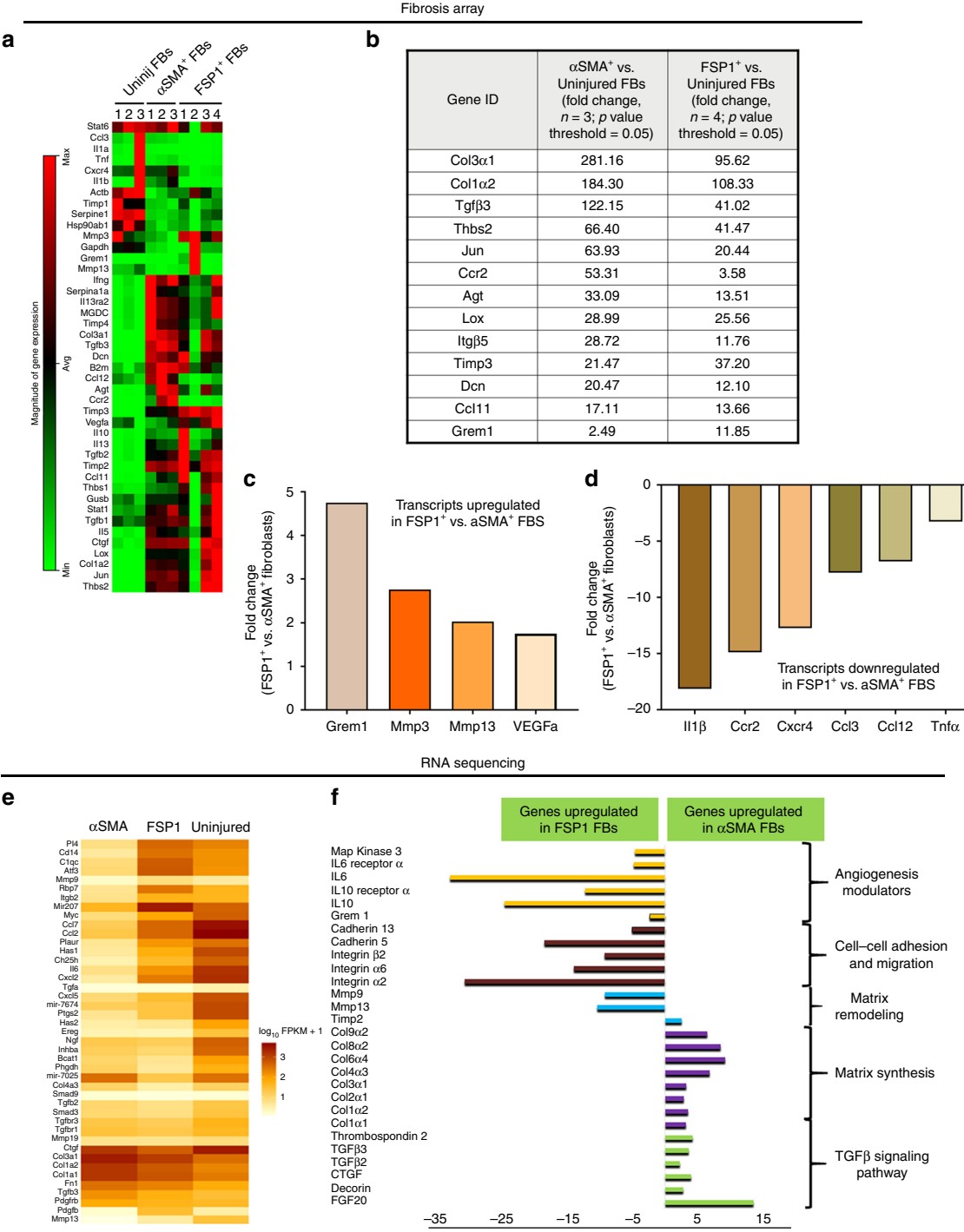

**Fig. 5** Distinct molecular signatures of uninjured, FSP1+, and αSMA+ fibroblasts identified by fibrosis array and RNA sequencing. **a** Heat map from fibrosis array representing magnitude of gene expression of genes expressed in freshly isolated uninjured, FSP1+, and αSMA+ fibroblasts (*n* = 3 for uninjured and αSMA+ fibroblasts, *n* = 4 for FSP1+ fibroblasts; cell isolation and FACS were performed at least three independent times. For each sorting, cells were isolated from pooled homogenates from three to four injured murine hearts 10 days post MI). FB = fibroblasts **b** Relative fold change of selected highest expressing genes identified by fibrosis array expressed by FSP1+ and αSMA+ fibroblasts compared with uninjured fibroblasts. **c** Relative fold change of upregulated genes expressed in FSP1+ fibroblasts compared with αSMA fibroblasts by fibrosis array. **d** Relative fold change of downregulated genes expressed by FSP1+ fibroblasts compared to αSMA+ fibroblasts by fibrosis array. **e** Hierarchical clustering (partial heat map representing log₁₀ values) of RNA transcript reads per kilobase per million mapped reads for genes with significant differential expression (fold difference > 2, *p* value < 0.05) between uninjured, FSP1+, and αSMA+ fibroblasts using CLC Genomics Workbench 8.0. **f** Relative fold change of genes representing key pathways identified by RNA sequencing in αSMA+ vs. FSP1+ fibroblasts. Genes downregulated in αSMA+ fibroblasts are mentioned as genes upregulated in FSP1+ fibroblasts. KEGG pathway analysis was performed on the RNA sequencing data on DAVID bioinformatics Resource 6.7 platform

downregulated in FSP1+ vs. αSMA+ fibroblasts. The list includes interleukin 1 beta (*IL1β*; −18.14 fold), C-C chemokine receptor type 2 (*Ccr2*; −14.89 fold), C-X-C chemokine receptor type 4 (*Cxcr4*; −12.74 fold), Chemokine (C-C motif) ligand 3 (*Ccl3*; −7.81 fold), chemokine (C-C motif) ligand 12 (*Ccl12*; −6.8 fold), and tumor necrosis factor (*Tnf*; −3.26 fold) (Fig. 5d). The fibrosis array results were confirmed by RNA-seq and distinct molecular signatures of all three fibroblast subtypes were identified (Fig. 5e). KEGG pathway grouping on the DAVID bioinformatics Resource 6.7 platform identified that the fibrillary collagen genes (*Col1a1, Col1a2, Col3a1*) and the constituents of the TGFβ signaling pathway were highly upregulated in αSMA+ fibroblasts as compared with FSP1+ fibroblasts, suggesting a distinct role of the αSMA+ population in injury-initiated fibrosis. By contrast, genes known to have a positive impact on angiogenesis (such as *Grem1, Il6, Il6r, Il10,* and *Il10r*), MMPs, and genes involved in adhesion and migration (*cadherin 5* and *13, integrin α2, α6,* and *β2*) were much more highly expressed in FSP1+ vs. αSMA+ fibroblasts (Fig. 5f). Cell metabolism, cell cycle, and RNA biogenesis pathways were not significantly changed between the groups and are not reported here. Two independent transcriptomic analyses of freshly sorted fibroblast subtypes suggested that FSP1+ fibroblasts contributed to a regulatory and pro-angiogenic role whereas αSMA+ fibroblasts were drivers of matrix synthesis and scar formation following injury.

**FSP1+ and αSMA+ fibroblasts respond to pro-fibrotic factors**. In response to injury, there is an activation of several growth factor and cytokine signaling, including WNT and TGFβ. Similar levels of TGFβ signaling pathway activation were identified in both αSMA+ and FSP1+ fibroblasts, assessed through SMAD2 phosphorylation (Supplementary Fig. 9). Both FSP1+ and αSMA+ fibroblasts responded to WNT3A treatment as assessed by pLRP6 immunoblot. However, the response was much more robust in αSMA+ vs. FSP1+ fibroblasts. In addition, COL1α1 protein levels were slightly higher in αSMA+ fibroblasts than FSP1+ fibroblasts in response to WNT3A treatment. These results suggested that these two populations exhibit distinct activation thresholds to common post-injury modulators.

**FSP1 expression is significantly induced by FGF2**. αSMA and FSP1 proteins are expressed in injury-induced fibroblasts. Although TGFβ has been a well-documented growth factor which induces αSMA expression in fibroblasts[26], little is known about the injury mediated triggers that induce FSP1 expression. We tested the effect of multiple growth factors such as TGFβ, WNT3A, PDGF, and FGF2 on their ability to induce FSP1 transcript expression in uninjured fibroblasts by semiquantitative real-time PCR. These growth factors are involved in the multi-faceted process of wound repair following injury[3,27–29]. Even though in the presence of WNT3A and PDGF there was a modest increase in FSP1 expression, a significant increase in FSP1 expression was identified only in the presence of FGF2 (Supplementary Fig. 10). Previous reports have identified WNT3A as a trigger for FSP1 expression[30]; however, our data suggest that FGF2 might be an important inducer for FSP1 expression in fibroblasts in the wound microenvironment. Regardless, these identifications do not rule out the complex interplay of multiple signaling cascades in regulating FSP1 expression in fibroblasts.

**FSP1+ fibroblasts exhibit a pro-angiogenic protein signature**. To validate the relatively higher pro-angiogenic transcriptional signature of FSP1+ fibroblasts, we used a multiplex panel of mouse angiogenesis proteome profiler antibody array (R & D) to measure 53 angiogenic analytes from FSP1+ vs. αSMA+

fibroblast lysates. Significantly increased expression of 14 out of 53 angiogenesis-related proteins were identified in FSP1+ fibroblasts including SDF1, CXCL16, Endoglin, Endothelin1, ADAMTS1, IGFBP-2 and -3, PDGF-AA, MMP-3, and Pentraxin-3 (Fig. 6a, Supplementary Fig. 11, and Supplementary Table 1). In contrast, only four angiogenesis-related proteins (DPPIV, osteopontin, serpin E1, and IP-10) were significantly upregulated in αSMA+ fibroblasts compared with FSP1+ fibroblasts (Fig. 6b, Supplementary Fig 11, and Supplementary Table 1). Several additional angiogenesis-related proteins exhibited a trend towards higher expression in FSP1+ vs. αSMA+ fibroblasts including VEGF, although the differences were not statistically significant. However, elevated expression of VEGF and GREM1 in FSP1+ fibroblasts were confirmed by an independent ELISA, which demonstrated significantly higher levels of secreted VEGF and GREM1 protein in conditioned media obtained from FSP1+ fibroblasts compared with αSMA+ fibroblasts (Fig. 6c). Moreover, elevated levels of GREM1 protein were also present in cell lysates of FSP1+ vs. SMA+ fibroblasts (Fig. 6c). In addition to *Grem1*, other pro-angiogenic genes such as *Vegfa, Vegfb, Angpt1,* and *Fgf1* were significantly elevated in FSP1 fibroblasts in comparison with uninjured and αSMA fibroblasts by semiquantitative real-time PCR (Fig. 6d). Altogether, these data indicated a much robust pro-angiogenic phenotype of FSP1+ fibroblasts as compared with αSMA+ fibroblasts.

**FSP1+ fibroblasts secretome is pro-angiogenic in vitro**. The pro-angiogenic gene and protein signature prompted us to assess the effect of FSP1+ fibroblast conditioned media (CM) on promoting human umbilical vein endothelial cells (HUVEC) proliferation measured by BrdU incorporation (Fig. 7c) as well as HUVECs assembly (Fig. 7a, b) into interconnected tube like structures on a growth factor reduced matrix (vascular assembly). Both assays are commonly used to demonstrate pro-angiogenic effects. FSP1+ fibroblast CM significantly increased HUVEC proliferation (~1.7 fold; $p < 0.0001$) and vascular assembly in HUVECs (137.3 vs. 53.88 branch points/field of view; $p = 0.0005$) compared with no treatment. HUVECs treated with complete endothelial cell growth media (Lonza) were used as positive control. Interestingly, FSP1 protein also increased HUVEC proliferation (~1.34 fold; $p = 0.0005$) and vascular assembly (89.14 vs. 53.88 branch points/field of view; $p = 0.0005$) compared with no treatment, albeit not at the same extent as induced by FSP1+ fibroblast CM. The CM generated from the uninjured fibroblasts and αSMA+ fibroblast had marginal effect on HUVEC proliferation (Fig. 7f) and vascular assembly (Fig. 7a, b). These results suggested an angiogenesis promoting phenotype of FSP1-expressing fibroblasts which could be partly attributed to FSP1 protein, as previously reported[31,32].

**FSP1+ fibroblasts are pro-angiogenic in vivo**. To test if FSP1+ fibroblasts serve as a superior pro-angiogenic subpopulation in vivo, we implanted polyvinyl alcohol (PVA) sponges containing distinct fibroblast subtypes to study the granulation tissue formation. Implantation of PVA sponge discs is a well-established model used to study granulation tissue formation in vivo[33]. Sponges were preloaded with freshly isolated and sorted FSP1+ (CD45−/CD31−) or αSMA+ cardiac fibroblasts prior to their subcutaneous implantation beneath the ventral panniculus carnosus in adult mice. The mice were sacrificed on day 14, a time point in which the majority of the granulation tissue is generated by implanted fibroblasts, not by infiltrating host fibroblasts. Sponges were fixed with formalin and embedded for histological analysis to quantify the amount and quality of granulation tissue, vascularity (anti-CD31), and matrix deposition (Masson's

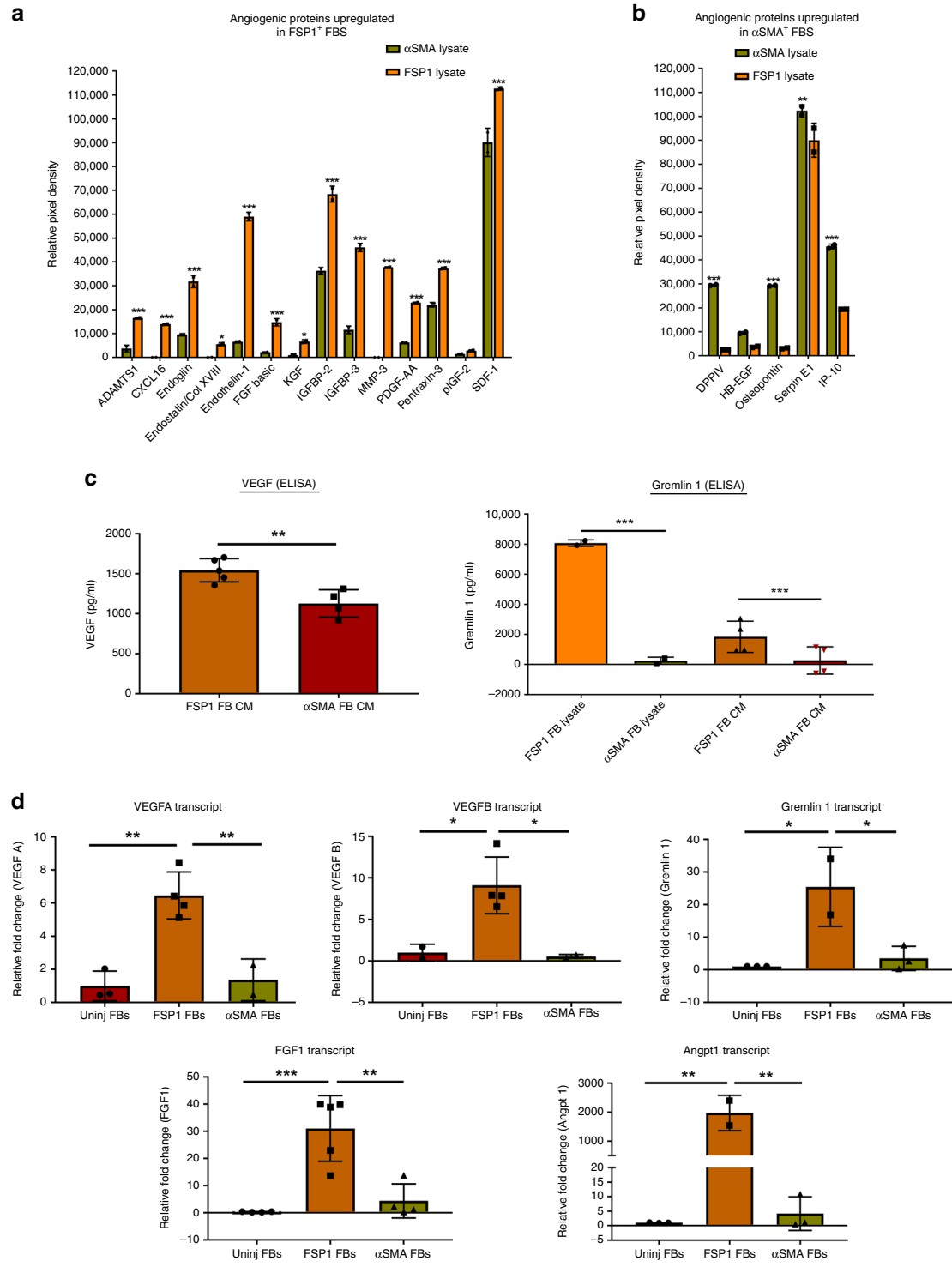

trichrome). Although we did not identify any statistically significant difference between the quality or extent of granulation tissue deposition, there was a striking increase in the percentage of CD31 stained vessels in sponges loaded with FSP1+ fibroblasts (6.817% ± 0.5496 SD; $p < 0.0001$) compared with αSMA+ fibroblast loaded sponges (1.78% ± 0.2463 SD; Fig. 7d, e). Notably, FSP1+ and αSMA+ fibroblasts, sorted for implantation, stained negative for endothelial cell markers (VWF and FLK-1), suggesting that the increased vasculature in FSP1+ fibroblast loaded sponges was host derived, and not derived from implanted FSP1+ or αSMA+ cells (Fig. 7g). These data have identified a significantly greater pro-angiogenic reparative role of FSP1+ than

αSMA+ fibroblasts during cardiac healing following injury both in vitro and in vivo.

## Discussion

Fibroblasts are recognized as a heterogeneous population of cells in several organ injury models[5,6,34–36]. Recently, high-throughput single cell transcriptomic studies have identified several fibroblast subtypes in murine kidney and heart as well as in human skin tissues[37–39]. However, a critical gap exists between recognizing the heterogeneity of fibroblast subtypes and identifying/understanding their functional differences and distinct roles on healing

**Fig. 6** FSP1[+] fibroblasts exhibit pro-angiogenic protein signature in vitro. **a** Relative pixel density of selected angiogenic proteins significantly high in FSP1[+] vs. αSMA[+] fibroblast lysates (P3–P5) identified using mouse angiogenesis proteome profiler array. *$p < 0.01$ and ***$p < 0.0001$ were calculated by two-way ANOVA using Bonferroni post-test, $n = 2$ technical replicates were performed; bar represents mean ± SD. **b** Relative pixel density of selected angiogenic proteins significantly high in αSMA[+] vs. FSP1[+] fibroblast lysates (P3–P5) identified using mouse angiogenesis proteome profiler array. **$p < 0.001$ and ***$p < 0.0001$ were calculated by two-way ANOVA using Bonferroni post-test, $n = 2$ technical replicates were performed; bar represents mean ± SD. **c** (left panel) Quantification of secreted vascular endothelial growth factor (VEGF) by Quantikine VEGF ELISA in the conditioned media (CM) obtained from FSP1[+] ($n = 5$) and αSMA[+] ($n = 4$) fibroblasts cultured for 72 h at 37 °C. **$p < 0.001$ was calculated by unpaired $t$ test; bar represents mean ± SD. (Right panel) Quantification of Gremlin 1 by ELISA in cell lysates ($n = 2$) and CM ($n = 4$) obtained from FSP1[+] and αSMA[+] fibroblasts cultured (P3–P5) for 72 h at 37 °C. ***$p < 0.0001$ was calculated by one-way ANOVA; bar represents mean ± SD. **d** Relative fold change of *Vegfa* ($n = 3$ for uninjured, $n = 4$ for FSP1[+], and $n = 2$ for αSMA[+] fibroblasts), *Vegfb* ($n = 2$ for uninjured, $n = 4$ for FSP1[+], and $n = 2$ for αSMA[+] fibroblasts), *Grem1* ($n = 3$ for uninjured, $n = 2$ for FSP1[+], and $n = 3$ for αSMA[+] fibroblasts), *Angpt1* (($n = 3$ for uninjured, $n = 2$ for FSP1[+], and $n = 3$ for αSMA[+] fibroblasts), and *Fgf1* ($n = 4$ for uninjured, $n = 5$ for FSP1[+], and $n = 4$ for αSMA[+] fibroblasts) transcripts measured by real-time RT-PCR in uninjured, FSP1[+], and αSMA[+] fibroblasts (P0–P5) . *$p < 0.01$, **$p < 0.001$, and ***$p < 0.0001$ was calculated by one-way ANOVA; bar represents mean ± SD

and fibrosis[7]. In fact, fibroblasts have been frequently studied as a homogenous entity when considering their functional roles in the injured heart as well as in other tissues[2,40,41]. Though in uninjured organs, Thy1[+] and Thy1[−] fibroblast subtypes have designated functional roles which includes differences in their ability to form focal adhesions, cytoskeletal organization and migration[42], in addition to differential expression of MHC class-II and CD40 expression, cyclooxygenase localization, and cytokine and prostaglandin production[35,43]. Thy1[+] and Thy1[−] fibroblasts have been identified in injured (fibrotic) lungs[44]; albeit without further functional characterization. Recently, bulk and single cell transcriptomics were utilized to predict the function of heterogeneous synovial fibroblast subtypes isolated from human rheumatoid arthritis patients[45]. Based on functional assessment, a potentially pathogenic CD34[−]/Thy1[+] fibroblast subtype was identified in vitro, the role of which needs to be assessed in vivo.

Our goal in the present study is to delineate the functional role of injury-induced activated two fibroblast subtypes in vitro as well as in vivo in order to better understand, target and mitigate fibrosis. Since post-injury-activated fibroblasts are both mediators of fibrosis and repair[2,3], understanding the functional differences in subpopulations may enable us to modulate detrimental fibrosis more precisely. The use of αSMA-GFP and FSP1-GFP mouse models have enabled us to FACS sort highly purified populations of distinct fibroblast subtypes, while carefully eliminating contaminating myeloid cells by gating where appropriate (i.e., FSP1-GFP). Subsequent immunophenotyping confirmed the absence of contaminating myeloid/hematopoietic and endothelial cell types from our isolated populations of both FSP1 and αSMA fibroblasts. We noted that one limitation of the αSMA-GFP model was that the transgene expression lagged behind the αSMA protein expression following MI, achieving a high degree of concordance (> 90%) at day 10 post injury. As a consequence, we used day 10 post MI for isolating αSMA fibroblasts.

While FSP1[+] and αSMA[+] fibroblasts have been reported as distinct fibroblast populations in injured tissues[11–13] and cancer[15], no reports to date have elucidated their functional and molecular differences. In this study we have comprehensively evaluated these phenotypes both temporally and in multiple mouse models of tissue injury and in human infarcts. We have confirmed prior studies that FSP1[+] and αSMA[+] fibroblast subtypes are indeed distinct populations but also demonstrated that they appeared at different times following injury. FSP1-expressing fibroblasts were present at the wound site earlier than αSMA-expressing cells in three different injury models. We have identified that the FSP1-expressing fibroblast subtype has a predominant role in promoting angiogenesis post injury via multiple approaches, including transcriptomic, phenotypic, molecular, and functional analysis.

The dynamic process of wound healing requires several continuous and overlapping phases which includes an early phase of cellular homeostasis, cell recruitment, and vascularization, and a later phase of re-epithelization and collagen deposition[46]. Fibroblasts, inclusive of all subtypes, have been implicated in immune cell recruitment and vessel formation as well as in collagen deposition and scar formation in both early and late wound healing[2,40,41,47]. However, discrete roles of particular subtypes in regulating these overlapping processes have not been delineated. αSMA myofibroblast activation during the later phase of wound healing has been linked to wound contraction and scar formation to accelerate their progression[40]. Onset of FSP1[+] fibroblasts during the early phase of wound healing is suggestive of their contribution to immune cell recruitment, cellular proliferation and angiogenesis. Indeed, our transcriptomic data identified the elevation of chemo-attractants and chemokines such as IL-10 and IL-6, as well as MMP-3 and MMP-13 in FSP1[+] fibroblasts compared with αSMA[+] fibroblasts. In addition to being involved in injury-induced leukocyte infiltration, the aforementioned cytokines and MMPs have also been implicated in fostering regenerative wound repair and angiogenesis[48–51]. Many landmark studies have shown that macrophages are critical players of wound healing and that they do so at least in part by promoting angiogenesis and collagen matrix deposition[52]. Significant work has been done to understand the crosstalk between activated fibroblasts and macrophages during wound repair[53]. However, the role of a specific fibroblast subtype in modulating the early phase of wound healing, which involves immune cell recruitment, has not been explored. Our identification of a sorted FSP1[+] fibroblast population with a specific gene signature suggests their involvement in leukocyte recruitment, angiogenesis promotion, and regenerative wound healing. Leukocyte crosstalk with unique fibroblast subtype(s) is an underexplored area and will be the focus of future studies.

αSMA-expressing post-injury-activated myofibroblasts are well recognized as a major contributor to matrix deposition, contraction, and scar formation following injury[2,5]. Conversely, FSP1-expressing fibroblasts, although accepted as post-injury-activated fibroblast, are less appreciated for their role in post-injury healing. Other cell types such as hematopoietic and endothelial cells also express the FSP1 protein following injury[8], creating ambiguity in assigning a specific role to FSP1-expressing fibroblasts. Due to this, FSP1-expressing cells are mostly considered to be of hematopoietic origin with very little attention to a subset of FSP1[+] cells that are fibroblasts[10]. In the present study, using BMT, we found that the majority of the FSP1[+] fibroblasts originated at the site of injury and are non-hematopoietic. Freshly sorted non-hematopoietic and non-endothelial FSP1[+] cells stained positive for fibroblast markers, including MEF-SK4, and

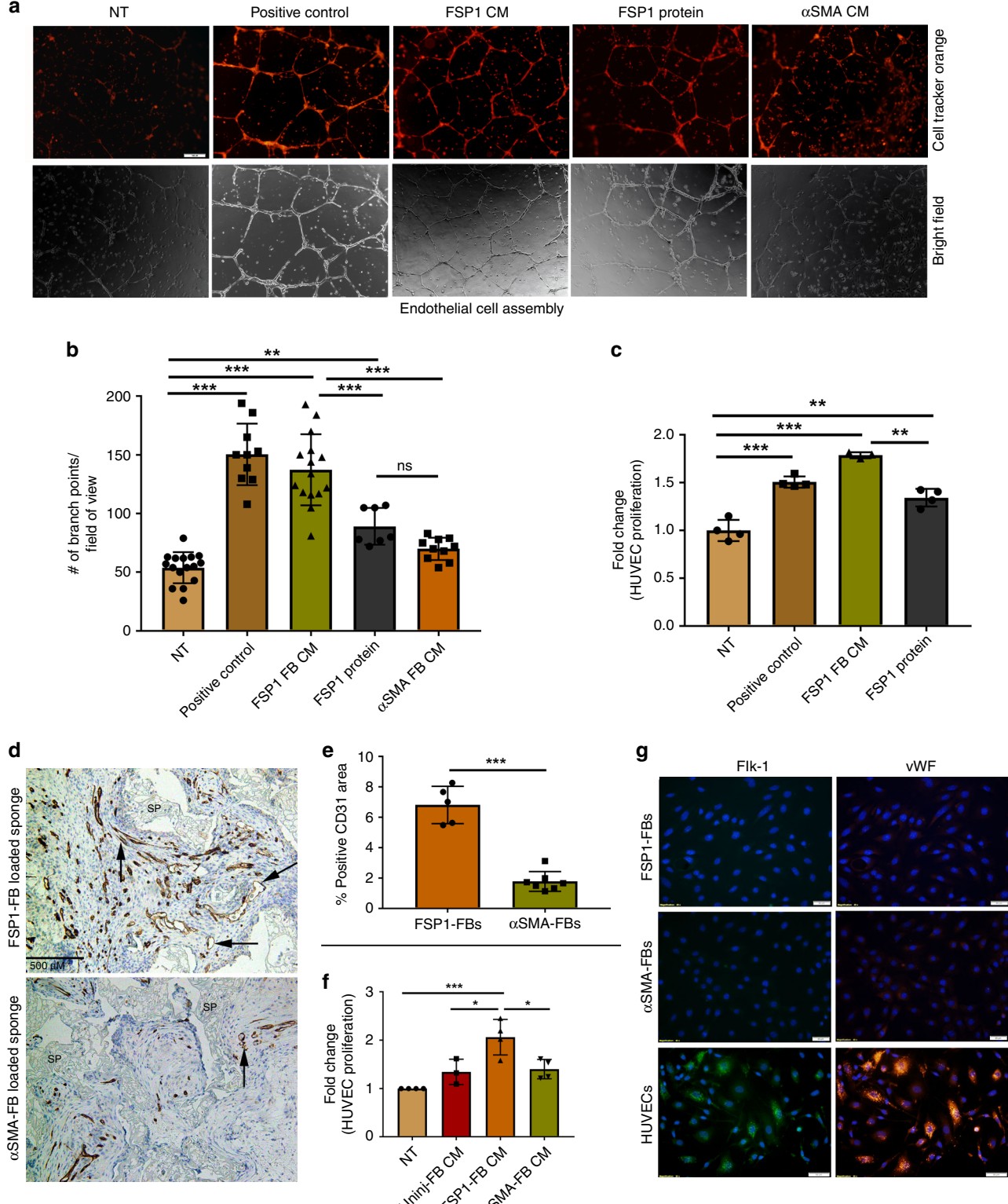

negative for pericyte marker AN2/NG2, were ultrastructurally similar to activated fibroblasts, and were able to contract collagen gel similar to uninjured and αSMA⁺ fibroblasts, confirming the fibroblastic phenotype of FSP1⁺ fibroblasts.

In addition, αSMA⁺ and FSP1⁺ fibroblast subtypes proliferated at a similar rate indicating fibroblast heterogeneity is not likely to modulate the post-injury fibroblast proliferation, consistent with work done by others[54]. It is possible that shared

molecular mechanism(s) stimulate the activation and proliferation of both fibroblast subtypes. Our observation that both αSMA⁺ and FSP1⁺ fibroblasts respond to pro-fibrotic proteins, TGFβ and WNT3A, supports this argument.

We found that both activated fibroblast subtypes express COL1A1 and periostin. Although, COL1A1 is known to be highly expressed by myofibroblasts, its presence is not unique to fibroblasts and myofibroblasts[55]. Conversely, lineage tracing studies in

**Fig. 7** FSP1[+] fibroblasts are pro-angiogenic in vitro and in vivo. **a** Representative images from an endothelial cell assembly assay using HUVECs treated with conditioned media (CM) collected from fibroblasts (P3–P5) or FSP1 recombinant protein. In the top row, the endothelial cells were stained with cell tracker orange and the second row contains corresponding bright field images (×20). Scale bar = 100 nM. **b** Quantification from the endothelial cell assembly assay based on the number of intersecting branch points per field of view (×10). Data represent an average of HUVEC cell assembly from $n = 16$ for NT, $n = 10$ for positive control, $n = 15$ for FSP1[+] FB CM, $n = 7$ for FSP1 protein, and $n = 10$ for αSMA[+] FB CM images from three independent biological replicates; bar represents mean ± SD. **c** BrdU proliferation assay performed on HUVECs (P5) in response to either FSP1[+] fibroblasts conditioned media (with 0.2% serum; $n = 3$) or FSP1 recombinant protein (10 nM; $n = 4$). HUVECs cultured in full serum (2%; $n = 4$) were used as positive control. **d** Representative images of FSP1[+] and αSMA[+] fibroblast soak loaded sponges stained with CD31 to analyze vascular density. SP = sponge matrix, arrows point at positive stain. Scale bar = 500 μM. **e** Vascular density graphed as percentage of immunopositive CD31 area/total tissue area in histologic sections from granulation tissue. Data represent averages of multiple 40× fields from unpaired samples ($n = 6$ for αSMA[+] fibroblast loaded sponges and $n = 5$ for FSP1[+] fibroblast loaded sponges). **f** BrdU proliferation assay performed on HUVECs (P5) in response to NT ($n = 4$), FSP1[+] ($n = 4$), αSMA[+] ($n = 4$), and uninjured ($n = 3$) fibroblast (P3–P5) conditioned media. **g** Immunofluorescence staining of GFP[+] FACS sorted fibroblasts (P3–P5) isolated from left ventricle of FSP1-GFP or αSMA-GFP mice indicate that FSP1[+] and αSMA[+] cell populations do not express endothelial markers such as vWF or Flk-1 (VEGFR2). HUVECs were used as positive control ($n = 3$). [ns]$p > 0.05$, *$p < 0.05$, **$p < 0.001$, and ***$p < 0.0001$ was calculated by one-way ANOVA, $n = 3$ experiments were performed; bar represents mean ± SD

the damaged heart have shown that periostin is expressed by a majority of myofibroblasts following MI but is not expressed to a similar extent by FSP1[+] fibroblasts[8,10]. The reason for this discrepancy could be due to the presence of other FSP1-expressing cell types, which may have masked the presence of FSP1-expressing fibroblasts. We excluded other cell subsets in our isolation to obtain an active periostin and MEF-SK4 expressing FSP1[+] fibroblast population. The presence and functional role of periostin in wound remodeling has been documented[56] and its presence in the two activated fibroblast subtypes suggests that both FSP1[+] and αSMA[+] fibroblasts actively participate in the wound repair and remodeling process. Whereas MEF-SK4 has been shown to be a reliable fibroblast marker irrespective of their activation state[23].

We took two approaches to identify the transcriptional profiles of these two fibroblast subsets. The fibrosis gene expression array provided a comprehensive and focused view on the gene signature variation involved in the fibrotic process whereas RNA sequencing presented a detailed transcriptional snapshot while confirming the fibrosis array data. Elevated expression of genes involved in matrix synthesis and the TGFβ signaling pathway have been known to be a hallmark of the myofibroblast phenotype[24,25,40]. The cardiac FSP1[+] fibroblast transcript signature has not been characterized in detail. Expression of matrix synthesis genes to a lesser extent in FSP1[+] fibroblasts than in αSMA[+] fibroblasts suggests that the FSP1[+] fibroblast lacks a strong fibrotic phenotype compared with αSMA[+] fibroblasts. This premise was supported by the identification that several genes that are involved in promoting fibrosis, such as *Il1b*, *Ccr2*, *Ccl12* (a ligand for *Ccr2*), *Cxcr4*, and *Tnf*, were downregulated in FSP1[+] fibroblasts in comparison with αSMA[+] fibroblasts favoring a predominant role of FSP1[+] fibroblasts in reparative vs. fibrotic wound healing[57–60].

Fibroblasts contribute to both physiological and pathological angiogenesis[61,62]. While fibroblast activation has been largely focused on the periostin and αSMA-expressing myofibroblast phenotype[2,5], FSP1-expressing activated fibroblasts were shown to support a pro-angiogenic microenvironment in breast cancer metastasis[63]. According to this report, secreted growth factors expressed by FSP1[+] cells and ECM proteins played key roles in metastatic colonization. This study supports our findings of the pro-angiogenic role of FSP1[+] fibroblasts during fibrotic wound healing. Interestingly, the role of FSP1 protein in promoting angiogenesis has been documented by others in tumor angiogenesis[31,32] but not in wound repair. However, our findings demonstrated that the pro-angiogenic effects of FSP1[+] fibroblasts observed in vivo and in vitro using their secretome (conditioned media) was more robust than that induced just by recombinant FSP1 protein. These observations suggest that the pro-angiogenic effect of FSP1-expressing fibroblast secretome was due to the combined effects of FSP1 protein itself as well as other factors. Angiogenesis requires the concerted action of multiple cytokines and growth factors[64] that were evident in FSP1[+] fibroblasts based upon the angiogenesis proteome array analysis. In addition to *Vegfa* and *Vegfb*, we identified other pro-angiogenic factors, including *Grem1*, *Angpt1*, and *Fgf1*, that were robustly expressed in FSP1[+] fibroblasts. This indicates a strong pro-angiogenic signature presented by FSP1-expressing fibroblasts. Although VEGFA, VEGFB, ANGIOPOETIN 1, and FGF1 proteins are hallmarks of angiogenesis[65,66], the significance of GREM1 in promoting angiogenesis has received attention by only a few elegant studies[67,68]. The orchestrated effort of multiple pro-angiogenic factors, including GREM1, in imparting a functional role to FSP1[+] fibroblasts needs to be explored in future studies. Altogether, both molecular and biochemical data in vitro as well as a wound healing model in vivo suggest a pro-angiogenic phenotype in FSP1[+] fibroblasts.

We identified two distinct activated fibroblast subtypes in several forms of tissue repair. The FSP1[+] fibroblast subset appears to play a significant role in promoting angiogenesis and possibly regenerative wound healing. The FSP1[+] population is distinct from the well-established, injury-activated αSMA[+] myofibroblasts. The heterogeneity of activated fibroblasts likely reflects the complex nature of organ repair and fibrosis, and it may also explain the failure to develop a robust anti-fibrotic therapy that overlooks such functional heterogeneity. Our studies serve to clarify the functional and molecular uniqueness of two distinct but prevalent post-injury fibroblast subtypes with the hope that greater understanding will lead to improved therapeutic targeting to mitigate fibrosis without altering functions necessary for wound healing.

## Methods

**Animals**. Wild-type C57Bl/6 (WT) mice were purchased from Jackson Laboratory (Bar Harbor, ME, http://www.jax.org). GFP-FSP1 and GFP-SMA mice were generous gifts from Drs. William E. Lawson (generated by Dr. Eric Neilson[20]) and Ivo Kalajzic[21], respectively. This study was carried out in strict accordance with the recommendations in the Guide for the Care and Use of Laboratory Animals of the National Institutes of Health. The Vanderbilt University Institutional Animal Care and Use Committee approved the protocol (Protocol number: M1600076-00). This study complies with all relevant ethical regulations. All experiments were performed using appropriate analgesics and anesthetics, and every effort was made to minimize pain/distress.

**Bone marrow transplantation (BMT)**. Unfractionated BM was obtained from C57Bl/6 mice[69]. FSP1-GFP mice were preconditioned with 10 Gy of γ-radiation from a 137Cs source before intraperitoneal injection with $5 \times 10^6$ nucleated donor

cells obtained from C57/BL6 mice. Recipient mice were given 2 months for recovery and engraftment post transplant before MI was performed.

**Myocardial infarction.** MI surgeries were performed at the Cardiovascular Physiology Core at Vanderbilt University Medical Center. For the MI model, mice were anesthetized with sodium pentathol (50 mg/kg) and endotracheal intubation was performed under direct laryngoscopy. Mice were ventilated with a small animal respirator (tidal volume = 1.0 ml, rate = 110 breaths/min). With the use of a surgical microscope, a left thoracotomy was performed. The fourth intercostal space was entered using scissors and blunt dissection. A 7-0 silk suture was placed through the myocardium into anterolateral LV wall (around the left anterior descending artery) and the artery ligated. Each heart was monitored with continuous EKG throughout the procedure to ensure successful infarction. The chest was closed in layers with 6-0 silk and the animal was gradually weaned from the respirator to avoid complicating pneumothorax. Intraperitoneal administration of buprenorphine (0.1 mg/kg) immediately following surgery and every 8–12 h for 72 h post surgery was used as analgesic. Animals were monitored closely for signs of distress and weight loss throughout the study period. Following study completion, or upon distress, animals were euthanized by overdose of isoflurane followed by cervical dislocation.

Human heart tissues were collected from cadavers without MI as well as from cadavers who passed away within 2 weeks post MI. Paraffin blocks of postmortem human heart tissue were obtained from the Vanderbilt Medical Center Tissue Repository through VUMC Cardiovascular Translational and Clinical Research (CLTCR) via IRB approval (Vanderbilt IRB No: 130881). Informed consent was obtained with consent for autopsy to use the postmortem tissues for research in a de-identified manner. Experiments with these tissues were performed in adherence to the relevant ethical guidelines. Patient de-identified paraffin embedded slides of human hearts were generated at Translational Pathology Shared Resource (TPSR) Vanderbilt University Medical Center.

**Cell culture.** Human umbilical vein endothelial cells (HUVECs) were purchased from Lonza (C2519A; pooled donor) and were maintained in EBM-2[TM] (Lonza; CC-3156) media supplemented with EGM-2[TM] Single Quots (Lonza; CC-4176), fungizone and antibiotics. Human retinal pericytes (Cell Systems ACBRI 183), generous gift from Dr. John Penn's lab, were maintained in DMEM supplemented with 10% FBS and EGM-2[TM] Single Quots. Primary murine pulmonary microvascular endothelial cells were isolated and generously shared by Dr. Jin Chen's laboratory[70]. Primary uninjured, FSP1[+] and αSMA[+] fibroblasts were isolated and sorted from murine hearts and cultured in DMEM-F12 with 10% FBS at 37 °C/5% $CO_2$. In some experiments, cells (P3–P5) were treated with TGFβ (10 ng/ml; R&D Systems: 240-B-010) or WNT3A (50 ng/ml; R&D Systems: 1324-WN) or PDGFβ (1:1000; Gibco: PHG004), or FGF-2 (25 ng/ml; Prospec: cyt-386) for 24 h.

**Primary fibroblast isolation.** Primary mouse cardiac fibroblasts were isolated according to previous protocol[10]. Uninjured C57Bl/6 mice were used to isolate uninjured fibroblasts whereas FSP1-GFP and αSMA-GFP mice were used to isolate and sort GFP-expressing fibroblasts 10 days following MI. In short, mice at least 12-weeks-old were euthanized by overdose of isoflurane followed by cervical dislocation. Heart tissue was dissected to isolate ventricular tissue, which was then placed into Kreba-Henseleit (KHB; Sigma; K3753a) buffer with 2.9 mM $CaCl_2$ and 24 mM $NaHCO_3$ and quickly minced. The minced tissue was then transferred to a digestion cocktail of 0.25 mg/mL Liberase Blendzyme 3 (Roche Applied Science), 20 U/mL DNase I (Sigma Aldrich), 10 mmol/L HEPES (Invitrogen), and 0.1% sodium azide in HBSS and stirred at 37 °C for 20 min. Cells collected after digestion were passed through 40 μm filter and centrifuged at 400 × g for 15 min. To remove red blood cells, cell pellet was resuspended in 1 ml of Red Blood Cell lysis buffer and incubated for 1 min. Following incubation, cells were washed with the KHB buffer and centrifuged at 400 × g for 15 min. Cells were resuspended in DMEM-F12 with 10% FBS. For uninjured fibroblast isolation, cells were plated into 6-well dish for separation of fibroblasts by selective adhesion for 4 h. Unattached and dead cells were washed with PBS and fresh media was added to the attached fibroblast for culture. For FSP1 and αSMA-expressing fibroblasts isolation, GFP-expressing cells were sorted by fluorescence assisted cell sorting (FACS) as described in the "Flow cytometry" section.

**Flow cytometry.** Isolated αSMA[+] or FSP1[+] fibroblasts were flow sorted for green fluorescent protein (GFP) 10 days following MI. Dead cells were excluded by either using 7AAD (Molecular Probes: A13010) or Ghost dye[TM] violet 510 (13-0870; Tonbo Biosciences). Unstained uninjured fibroblasts were used to set the background signal in the GFP channel post compensation. A gate was drawn to capture all cells producing a larger voltage pulse area than the established background in the GFP channel. GFP[+] FSP1 cells were gated with APC-CD45 (1:200; BD Bioscience: 559864) and PE-CD31 (1:200; BD Bioscience: 553373) antibodies to exclude FSP1-expressing endothelial and hematopoietic cells. For αSMA-GFP mice, live GFP-positive cells were selected by gating for 7-AAD or Ghost dye[TM] violet 510-negative, GFP-positive cells.

For analyses, cells were incubated with Fc blocker (purified anti-mouse CD16/CD32 2.4G2; 70-0161; Tonbo Biosciences) prior to staining with the following

antibodies: anti-feeder cells-APC (MEF-SK4) (1:10; Miltenyi Biotec: 130-102-900), anti-AN2/NG2 (1:11; Miltenyi Biotec: 130-097-455), APC-CD45, or PE-CD31 for 30 min 7AAD (1:1000; Molecular Probes: A13010) or Ghost dye[TM] violet 510 (1:1000; 13-0870; Tonbo Biosciences) were used to exclude dead cells. For unconjugated antibodies, donkey anti-rat alexa fluor 647 (1:100; Abcam: ab175670) secondary antibody was used. Flow cytometry analyses were performed in the VMC Flow Cytometry Shared Resource.

**Concentration of condition media.** In total, $1 \times 10^6$ FSP1[+], αSMA[+], or uninjured primary fibroblast cells (P3–P5) were plated in 10 cm[2] dishes in DMEM-F12 containing 10% FBS. After cells attached, media was exchanged for DMEM F12 with 0.5% FBS or alternatively with EBM-2 containing 0.2% FBS and incubated at 37 °C for 72 h. Conditioned media (CM) was collected and then concentrated by centrifugation at 3500 × g with the help of Amicon Ultra centrifugal filter units (Millipore: UFC900324) for 30 min. Protein concentration was assessed by bicinchoninic acid assay (BCA assay). CM generated in EBM-2 was used to treat HUVECs for proliferation and endothelial cell assembly assays.

**Enzyme linked immunosorbent assay (ELISA).** The Quantikine mouse VEGF ELISA kit (R&D Systems; #MMV00) and Gremlin-1 ELISA kit (Raybiotech # ELM-GREM1) were utilized to quantify VEGF or Gremlin-1 in conditioned media (100 μg/cell type) or cell extracts (50 μg/cell type) of fibroblast subtypes (P3–P5). All assays were performed in triplicate according to each manufacturer's instructions.

**Angiogenesis protein array.** The Proteome Profiler Mouse Angiogenesis Array Kit (# ARY015; R & D) was used to assess the relative levels of 53 mouse angiogenesis-related proteins in 300 μg of FSP1[+] and αSMA[+] cell lysates according to manufacturer's instructions. Angiogenesis Array is a membrane-based sandwich immunoassay. The cells were cultured for 72 h in 0.5% FBS prior to protein collection. Briefly, a cocktail of biotinylated detection antibodies was mixed with cell lysates. The array membrane was then treated with the cell lysate and antibody mixture overnight at 4 °C, washed, and then incubated with streptavidin-HRP. The array membrane is spotted with capture antibodies to specific target proteins. The captured proteins were visualized using chemiluminescence.

**In vitro vascular assembly assay.** HUVECS (P5) were serum starved with EBM-2 containing 0.2% FBS overnight, then treated with CellTracker[TM] Orange CMRA Dye (C34551; Life Technologies) for 1 h at 37 °C. Twenty-four well plates were coated with 150 μl of matrigel (Corning: CB-40230) for 1 h at 37 °C. Following incubation, $3 \times 10^4$ HUVECS were seeded into each well with conditioned media collected from either FSP[+], αSMA[+], or uninjured fibroblasts. After 16 h, images (×4 and ×10) were taken using an Olympus CK40 inverted microscope through an Optronics DEI-750C charge-coupled device video camera using Cellsens Dimension software. Vascular assembly was assessed by counting the number of intersecting branches[70].

**Cell proliferation assays.** Cell proliferation was assessed by 5-bromo-2′-deoxyuridine (BrdU) cell proliferation assay (Calbiochem, Gibbstown, NJ). Following serum starvation, 2000 HUVECs (p5) or 1000 fibroblasts (P3–P5) were seeded per well in a 96-well plate. Following overnight attachment of cells to plate, media was replaced with fibroblast conditioned media (FSP1[+], αSMA[+], and uninjured). In some experiments, HUVECs were treated with 10 nM recombinant mouse S100A4 protein (R&D Systems: 4138-S4-050). BrdU incorporation was assessed by measured absorbance at dual wavelength of 450/595 nm following manufacture's protocol.

**Repair/granulation tissue stimulation.** PVA sponges (Medtronics Merocel (4 mg, 2 mm height, 4 mm diameter)) were soak loaded with freshly sorted $7.5 \times 10^5$ primary FSP1[+] or αSMA[+] fibroblasts in 20 μl of sterile PBS and were implanted in ventral subcutaneous pockets of adult C57Bl/6 mice. Mice were taken down at day 14 after implantation to remove the sponges. The sponges were cut in half and fixed in 10% neutral buffered formalin for 24 h, embedded cut side down in paraffin and sectioned for immunostaining.

**Histochemistry and morphometry.** Sponges embedded in paraffin were sectioned into 5-micron-thick transverse sections for histology. Vanderbilt Translational Pathology Shared Resource performed the immunohistochemistry for CD31 to analyze vascularity with anti-CD31/platelet endothelial cell adhesion molecule-1 (PECAM-1; clone 557355, PharMingen)[69]. Images of CD31 stained sections were obtained with a CoolSNAP Hq CCD camera (Photometrics). Five images from each section were acquired at defined magnification (×40) for vascular density. Each field was quantified using Image J (NIH) by outlining tissue and calculating percentage CD31[+] area per field.

**Immunofluorescence and immunohistochemistry.** Immunofluorescence staining was performed on cells as well as on paraffin embedded tissues. Primary fibroblasts

(P0–P1) were seeded and cultured on coverslips until 80% confluent prior to immunostaining. In some cases, uninjured, αSMA⁺, and FSP1⁺ fibroblasts were treated with 10 ng/ml TGFβ for 72 h prior to fixation and immunostaining. Sorted primary GFP⁺/CD45⁺ and GFP⁺/CD31⁺ cells were concentrated on a coverslip by cytospin for immunostaining. Cells were fixed with cold acetone for 15 min and washed three times with PBS. For tissue staining, slides were deparaffinized and hydrated through a series of xylene and ethanol baths. Heat-mediated antigen retrieval was performed by boiling in citrate buffer (pH 6). Slides/coverslips were blocked in 10% goat serum and incubated with primary antibodies overnight. Secondary antibody incubation was performed for 2 h after three washes with PBS. The slides were counterstained and mounted using DAPI in slow-fade mounting media (Invitrogen: P36931). The following primary antibodies were used: Fibroblast-specific protein 1 (FSP1) (1:100; Millipore: 07-2274), α-smooth muscle actin (αSMA) (1:1000; Sigma: A2547), COL1α1 (1:1000; MD Bioproducts: 203002), Periostin (1:100; Santa Cruz: SC67233), Vimentin (1:200; Sigma: V2258), CD31 (1:250; BD Biosciences), CD45 (1:250; BD Biosciences: 553076), AN2/NG2 (1:100; Invitrogen: MA5-24247), VWF Polyclonal Antibody (Thermo-Fisher; Cat No: PA5-16634), and VEGFR2 (Flk1) SINGLE CHAIN ANTIBODY (Fitzgerald; Cat No: 10R-V106A). The secondary antibodies used were: conjugated goat anti-mouse Alexa Fluor 488 (Molecular Probes: A11029) or goat anti-rabbit-FITC or goat anti-rabbit-Cy3 (Jackson Immunoresearch Laboratories: 711-165-152 or Southern Biotech: 4050-02), goat anti-rat Alexa Fluor 488 or goat anti-rat Alexa Fluor 647 (Molecular Probes: A11006 or; Thermo-Fisher: A21247), or goat anti-chicken-FITC (Jackson Immunoresearch Laboratories: 103-095-155). For confocal analysis, LSM510 (Zeiss) microscope was used to capture 1 μm optical slices (z stack); the images were analyzed using Fiji software. Cells (P0–P1) were used for electron microscopy and pictures were obtained using a Quanta 250 Environmental Electron Microscope.

**Collagen contraction assay**. For the gel contraction assay 900 μl of collagen solution (1.67 mg/mL Type 1 rat collagen, 20 mM HEPES, 44 mM NaHCO₃, 1× DMEM) was mixed with 100 μl cells ($3.3 \times 10^5$ cells/mL) (P3–P5) for a final concentration of 1.5 mg/mL collagen. The cell/collagen mix was plated in 48-well plate (300 μl, $10^5$ cells/well) and incubated for 15–20 min at 37 °C until gelled. The gel was separated from the walls of the well with the help of a 30G needle. DMEM (600 μl/well) supplemented with 10% FBS and 1 ng/mL TGFβ (R&D Systems, #76666-MB) was added on top of the gels. Plates were scanned at 0, 24, and 48 h.

**Western blot analysis**. FSP1⁺ and αSMA⁺ fibroblasts (P3–P5) were serum starved overnight and seeded in DMEM-F12 with 10% FBS at a density of $5 \times 10^5$ cells/well in a six-well plate. Cells were washed with PBS and lysed using RIPA buffer supplemented with protease and phosphatase inhibitors (Roche). Cell lysates were rocked for 30 min at 4 °C then centrifuged at $10,000 \times g$ for 15 min to collect the supernatant. Protein concentrations were determined using a bicinchoninic acid (BCA) kit (Thermo Scientific). Fifty micrograms of protein was denatured at 95 °C for 10 min in SDS sample buffer before being resolved on SDS-PAGE electrophoresis and transferred onto a nitrocellulose membrane (PerkinElmer). Blots were blocked in 5% nonfat milk and probed with antibodies against collagen type I (MD Bioproducts, #203002), pSMAD2 (Cell Signaling, #3108L), SMAD2/3 (Cell Signaling, #5678S), LRP6 (Santa Cruz Biotechnology, SC-25317), pLRP6 (Cell Signaling, 2568S), or β-actin (Sigma Aldrich, #A5441). Species-specific secondary antibodies were used and chemiluminescence (NEL104, PerkinElmer, Waltham, MA) was detected by autoradiography.

**Semiquantitative real-time PCR**. Cells (P3–P5) were snap frozen in Trizol Reagent (Invitrogen, 15596026). Following RNA isolation with the Trizol Reagent, first strand DNA synthesis was performed with 1 μg RNA using iScript cDNA synthesis kit (Bio-Rad 170-8890) and semiquantitative real-time PCR was performed. Each reaction was normalized against 18S. Primers used are listed in Supplementary Table 2 (Table S2).

**RNA sequencing**. Fibroblasts were isolated and sorted—described in "Primary fibroblast isolation" section—from uninjured, FSP1-GFP and αSMA-GFP mice. Uninjured FBs were isolated from uninfarcted C57Bl/6 mice[10]. Enzyme digested cells were seeded on tissue culture plate for 4 h. Unattached cells were washed, and plastic-adherent FBs were collected for RNA extraction. FSP1-GFP and αSMA-GFP fibroblasts were sorted by FACS as described earlier. RNA was prepared from freshly sorted fibroblasts with the help of Ambion RNAqueous™-Micro Total RNA Isolation Kit (Life Technologies).

Purified RNA was sent to Vanderbilt sequencing core (VANTAGE), paired end sequencing was performed at 75X depth. Raw data (FASTQ files) were uploaded to CLC Genomic Workbench 8 (Qiagen) and data quality assessment performed. Sequences were aligned to the Grmc38 platform of the *Mus musculus* genome using Tuxedo Suite, followed by normalization (RPKM). Data were annotated using Ensemble Release 83 and quantified at the gene level using CuffDiff. Since replicates were not use, proportional-based statistical analysis was performed using Baggerly's *t*-test with multiple hypothesis correction. Genes with fold difference of 2 and *p* value < 0.05 were retained for functional analysis. DAVID bioinformatics

Resource 6.7 platform was used for KEGG pathway analysis. The RNA-seq data are available at the NCBI GEO database with the accession number GSE130062.

**Fibrosis array**. Fibrosis array was performed on freshly isolated and sorted uninjured, FSP1⁺ and αSMA⁺ fibroblasts. RNA was prepared from freshly sorted fibroblasts with the help of Ambion RNAqueous™-Micro Micro Total RNA Isolation Kit (Life Technologies). The first strand cDNA was synthesized with 500 ng of total RNA with the help of RT² First Strand Kit as per manufacturer's instructions (Qiagen, SABiosciences Corp., Frederick, USA). Samples were diluted with RT² SYBR Green qPCR master mix (SABiosciences Corp.) and pipetted into 96-well Mouse Fibrosis PCR array plates (SABiosciences Corp., Frederick, USA) to perform real time PCR quantification. The assay was performed on a CFX96 Real Time PCR Detection System (Bio-Rad). Each 96-well array plate consists of a panel of 84 genes involved in cardiac fibrosis in addition to house keeping genes and internal controls. Raw data were exported into an excel sheet and analyzed using the integrated web-based automated software available through SABiosciences for PCR Array Data Analysis. Relative gene expression and fold change was calculated by comparative threshold cycle (ΔΔCt) method. A two fold or greater change (*p* value cutoff < 0.05) in gene expression was considered significant.

**Statistical analysis**. Statistical significance calculations were performed using unpaired *t* test, one-way or two-way analysis of variances (ANOVA) with Bonferroni correction for multiple-group comparisons using the Prism 7.02 software (Graph-pad prism). $p \le 0.05$ was considered significant.

## Data availiblity

The RNA-seq data are available at the NCBI GEO database with the accession number GSE130062. The data from the angiogenesis array are deposited on figshare and can be accessed at https://figshare.com/s/1e573d8d31707884f2b8. All other data supporting the finding of this study are available from the authors on request.

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

## Acknowledgements

The authors want to thank Drs. Jeffrey M. Davidson and Hind Lal for careful reading of the manuscript and their critical feedback. We also want to thank David K. Flaherty from Vanderbilt Flow Cytometry Shared Resource and Dr. Shan Wang from Division of Rheumatology and Immunology at the Vanderbilt University Medical Center for their input in flow cytometry data analysis and presentation. We thank Drs. William E. Lawson and Eric Neilson for providing FSP1-GFP mice and Ivo Kalajzic for the gift of αSMA-GFP mice. Research reported in this publication was supported by the National Institute of General Medical Sciences of the National Institutes of Health (NIH) under

Award Number R01GM118300 (S.S.), National Institute of Biomedical Imaging and Bioengineering of the NIH under Award Number R21EB019509 (P.P.Y.), National Blood Foundation Grant (S.S.), and Scientist Development Grant of the American Heart Association under Award Number 17SDG33630187 (S.S.). Confocal imaging and electron microscopy analyses were performed in part using the VU Cell Imaging Shared Resource (supported by NIH grants CA68485, DK20593, DK58404, DK59637, and EY08126). Flow cytometry analyses were performed at the VMC Flow Cytometry Shared Resource which is supported by the Vanderbilt Ingram Cancer Center (P30 CA68485) and the Vanderbilt Digestive Disease Research Center (DK058404). The Vanderbilt VANTAGE Core provided technical assistance for RNA sequencing work. VANTAGE is supported in part by CTSA Grant (5UL1 RR024975-03), the Vanderbilt Ingram Cancer Center (P30 CA68485), the Vanderbilt Vision Center (P30 EY08126), and NIH/NCRR (G20 RR030956).

## Author contributions

P.P.Y. and S.S. designed the experiments. S.S., S.M., and L.W. performed the experiments. P.P.Y., S.S., and S.M. analyzed the data and interpreted the results and wrote the paper.

## Additional information

**Competing interests:** The authors declare no competing interests.

