## [Peer Review File · Nature Communications]

Reviewers' comments:

Reviewer #1 (Remarks to the Author):

The study has an interesting and novel message, suggesting that the FSP-1+ subpopulation of fibroblasts is distinct from activated myofibroblasts, and may promote angiogenesis. Although this cell biological concept is novel, support for the angiogenic/reparative function of these cells is associative. The following major concerns need to be addressed:

Major comments:

1. There is a need to identify the FSP1+/CD31-/CD45- cells as fibroblasts. The current exclusion criteria used by the authors are not sufficient. For example, these cells could represent pericytes, or other mesenchymal cells without characteristics of fibroblasts. The authors could perform combinations of the following experiments: a) exclude a pericyte phenotype using markers for pericytes (PDGFRbeta, NG2, etc.) and absence of perivascular location in vivo, c) document expression of ECM proteins characteristic of fibroblasts (collagens, FN), or fibroblast markers (eg PDGFRalpha) c) examine ultrastructural morphology of these cells to document fibroblast-like characteristics.

2. The angiogenic profile of the FSP1+ cells needs to be better documented. The authors use PCR and protein arrays to imply angiogenic effects of fibroblasts. However, the genes listed are not necessarily angiogenic. For example in page 11, it is stated that "increased expression of 14 out of 53 pro-angiogenic proteins was identified in FSP-1+ fibroblasts." As is often the case, some of the proteins listed by the manufacturer as "angiogenic" are angiostatic (TSP2 is an example). Other proteins have context-dependent effects. In page 10, the list of "angiogenic genes" includes IL-6, IL-6R, IL10, IL10R etc. These cytokines and cytokine receptors cannot be considered "angiogenic", but had a wide range of actions on inflammatory cells, fibroblasts, endothelial cells, etc. Functional assays should be used to support the angiogenic phenotype (preferably more than one). Moreover, when attempting to identify mediators that may be responsible for angiogenic actions, the authors should assess genes/proteins with established credentials as regulators of the angiogenic response (VEGFs, angiopoietins, FGFs etc.)

3. Does FSP1 mediate angiogenic actions, or is it simply a marker? This could be tested, at least in vitro using FSP1 knockdown approaches in a fibroblast facilitated angiogenesis assay. If FSP1 expression is not necessary for the angiogenic properties of these cells, the authors should test the role of relevant FSP-1 cell-derived mediators in vitro.

4. There are no FSP1+ fibroblasts in normal hearts, but there is a significant increase in their numbers following infarction. Are these cells derived from resident fibroblasts? If so, which signals upregulated FSP1 expression in fibroblasts? The mechanism of activation likely does not involve TGF-beta (considering that this mediator promotes myofibroblast conversion). Limited in vitro work may be sufficient to address this concern.

5. The proposed reparative role of FSP1+ fibroblasts in cardiac healing is interesting, but is not documented. Considering the challenges in demonstrating such an effect in vivo, the authors should recognize this limitation and tone down related statements.

Reviewer #2 (Remarks to the Author):

Saraswati et al

Fibrosis is a significant problem associated with tissue dysfunction. Until recently the fibroblast population has been difficult to study due to a paucity of adequate markers. In this manuscript the authors use two distinct transgenic mouse lines to investigate phenotypic differences in fibroblasts. In their results the authors suggest that injury induced fibroblasts that express Fibroblast specific protein 1 have different properties than fibroblasts expressing α SMA. A primary distinction is that FSP1 fibroblasts are pro-angiogenic while α SMA expressing cells are not.

Major comments

The authors provide an interesting perspective on fibroblast subpopulations. They provide data suggesting that FSP1 positive cells are characteristically different from the typical myofibroblast α SMA positive cell. One important detail is that the authors need to be sure that there are no contaminating cells in their sorted population. While the authors demonstrate little contamination in figure 2, further validation should be used to demonstrate that this is true for all of the populations of cells with every sorting. Possibilities would be qPCR for CD31, PECAM, CD45, and other immune cell markers through out the experiments. Additionally, the materials and methods state that primary fibroblasts P0-P5 were seeded. For each experiment they should verify the passage of fibroblast used as it in most cases they state freshly isolated so it is unclear when the

passaged cells are used. In addition, validation of findings should be shown by additional means as suggested below.

In addition there are a few additional points that should be addressed:

The authors provide statistical evaluation of their data, but it is often difficult to discern how n values are counted. For example, the protein array had SD listed, but the authors state that there is an n of 2. Are the stats calculated by n of two in duplicate or are there only two data points being used for each fibroblast population. The authors should clearly state if the n value is a biologic replicate or not. More details on reproducibility would strengthen the manuscript.

The separation of GFP positive and negative cells is not clear from the flow diagrams. How did the investigators determine what population to identify as positive? Did the investigators include a live dead stain in their flow analysis? Also the authors should describe how the images in 2D were obtained using HRP conjugated antibodies. The cultures appear surprisingly homogeneous in figure 2D. Are these freshly plated cells from the flow plots after just 4 hours?

The authors should include information regarding what passage of cells were used for each experiment.

It would be informative for the authors to report % of fibroblasts using the respective transgenic markers in addition to staining for PDGFRa and mEF-SK4 antibody by flow cytometry two markers that have been reported previously for cardiac fibroblasts to determine if these two populations constitute the majority of cardiac fibroblasts.

As the aSMA GFP line has not been extensively described in the heart. The authors should perform some characterization of the GFP expression similar to what was performed for figure 1 and aSMA staining. What percent of aSMA cells are GFP positive under these various conditions? Are GFP expression and aSMA protein synonymous?

For all uninjured experiments is based on what is known about aSMA expression, it seems like the authors are comparing VSMC/pericytes to FSP1+ cells rather than uninjured fibroblasts. The authors need to address why the aSMA population in an uninjured heart would be considered a fibroblast.

QPCR for aSMA would support the immunofluorescence assay in fig 3 and provide quantitative data. Also the authors should add aSMA fibroblasts to the experiment.

It is odd that the authors did not validate the expression of proteins observed in the protein array. They should consider verifying 1-2 of the proteins that are most differentially expressed in 5A and B.

There are also some discrepancies in data for example Fig 4 shows that thrombospondin 2 RNA is upregulated in aSMA fibroblasts while thrombospondin 2 protein is predominantly in the FSP1 protein array. How do the authors explain these contradictions?

Minor comments

As it is very germane to this manuscript primary references should be provided on page 3 line 70 rather than a review.

The table does not clearly label the columns from the FSP1+ cells and the aSMA+ cells.

Typo known page 4 line 92

Reviewer #3 (Remarks to the Author):

This is a very elegant and comprehensive study of two distinct fibroblast populations in the heart. The authors make a compelling case for the distinctness of the two populations and their unique transcription profiles. I have a few comments about the data.

1. the focus on pro-angiogenic role of the FSP+ population is unwarranted and only diminishes the paper. First of all, any cell(s) expressing lots of VEGF will be "pro-angiogenic". So the entire data set devoted to the "pro-angiogenic" effect is largely meaningless. You don't need to prove that VEGF induces angiogenesis. Second, if the authors truly want to make this point, they need to knockout this entire population in the mouse and describe the phenotype. The existence of two distinct sets of fibroblasts is a much bigger story and should be the focus.

2. It would be interesting to know how uniform these populations are. single cell RNA-seq could provide major biological insights here

3. One wishes for a bit of insight into the origin of these populations, what drives their existence/expression profile differences and their biological roles.

Reviewers' comments:

Reviewer #1 (Remarks to the Author):

The study has an interesting and novel message, suggesting that the FSP1+ subpopulation of fibroblasts is distinct from activated myofibroblasts, and may promote angiogenesis. Although this cell biological concept is novel, support for the angiogenic/reparative function of these cells is associative. The following major concerns need to be addressed:

Major comments:

1. There is a need to identify the FSP1+/CD31-/CD45- cells as fibroblasts. The current exclusion criteria used by the authors are not sufficient. For example, these cells could represent pericytes, or other mesenchymal cells without characteristics of fibroblasts. The authors could perform combinations of the following experiments: a) exclude a pericyte phenotype using markers for pericytes (PDGFRbeta, NG2, etc.) and absence of perivascular location in vivo, c) document expression of ECM proteins characteristic of fibroblasts (collagens, FN), or fibroblast markers (eg PDGFRalpha) c) examine ultrastructural morphology of these cells to document fibroblast-like characteristics.

The authors agree to the reviewer's suggestions. We have addressed the issues raised by the reviewer in the following ways:

1. Electron microscopy was performed on freshly isolated uninjured, FSP1, and α SMA positive fibroblasts (**Please see new Fig. S4A and text on page 8**). Primary microvascular mouse endothelial cells were used as control. All three fibroblast cell types showed similarity in ultrastructural organelles. α SMA and FSP1 fibroblasts demonstrated abundance of rough endoplasmic reticulum, mitochondria, and golgi apparatus as shown by others for myofibroblasts (Brian Eyden, 2009, Ultrastructural Pathology). By contrast, endothelial cells showed fewer organelles and a dense cytoplasm.
2. Flow cytometric analysis of MEF-SK4, reported to be expressed on all fibroblasts, demonstrated expression on uninjured, FSP1 and α SMA cells (**Please see revised Fig. 2E and text on page 8**).

3. Real time RT-PCR of *Col1a1*, *Col3*, *Fibronectin*, and *Pdgfra* transcripts, each known to be expressed in fibroblasts, were expressed in both FSP1 and α SMA -positive cells isolated from the heart (**Please see new Fig. S4B and text on page 8**).
4. Evidence that FSP1 protein expression does not co-localize with a pericyte marker NG2 in infarcted heart using confocal imaging (z stack included) (**Please see new Fig. 3C and text on page 8**).
5. Flow cytometric analysis and real time RTPCR of freshly sorted GFP-FSP1 fibroblasts showed absence of pericyte marker NG2. Human retinal pericyte and mouse MSCs were used as positive controls (**Please see new Fig. 3A and B and text on page 8**).

Together, these data provide further support that both post-injury populations isolated from the heart in this study are fibroblasts by additional phenotypic and molecular characterizations and that FSP1 cells are not pericytes.

2. The angiogenic profile of the FSP1+ cells needs to be better documented. The authors use PCR and protein arrays to imply angiogenic effects of fibroblasts. However, the genes listed are not necessarily angiogenic. For example in page 11, it is stated that “increased expression of 14 out of 53 pro-angiogenic proteins was identified in FSP-1+ fibroblasts.” As is often the case, some of the proteins listed by the manufacturer as “angiogenic” are angiostatic (TSP2 is an example). Other proteins have context-dependent effects. In page 10, the list of “angiogenic genes” includes IL-6, IL-6R, IL10, IL10R etc. These cytokines and cytokine receptors cannot be considered “angiogenic”, but had a wide range of actions on inflammatory cells, fibroblasts, endothelial cells, etc. Functional assays should be used to support the angiogenic phenotype (preferably more than one). Moreover, when attempting to identify mediators that may be responsible for angiogenic actions, the authors should assess genes/proteins with established credentials as regulators of the angiogenic response (VEGFs, angiopoietins, FGFs etc.)

The authors agree that the use of word pro-angiogenic could be misleading. We have changed the language in the paper to describe the proteins identified by protein array as angiogenesis-related proteins and not pro-angiogenic proteins (**See text on page 12**). Additionally, we agree that some of the genes that were identified in the genomic analysis such as IL-6 and IL-10 and their receptors indirectly impact angiogenesis. In the revised manuscript we note that multiple studies have demonstrated their indirect effect on promoting vessel formation (**See text on page 10 and 17**). To strengthen our findings we have also presented gene expression profile of other more commonly associated regulators of angiogenesis (*Vegfa* and *Vegfb*, *Gremlin1*, *Angiopoetin 1*, *Fgf1*) in the revised manuscript (**Please see revised Fig. 6D and text on page 13**).

3. Does FSP1 mediate angiogenic actions, or is it simply a marker? This could be tested, at least in vitro using FSP1 knockdown approaches in a fibroblast facilitated angiogenesis assay. If FSP1 expression is not necessary for the angiogenic properties of these cells, the authors should test the role of relevant FSP-1 cell-derived mediators in vitro.

In this study, we identified a major angiogenic role of an injury-induced fibroblast subtype which expresses FSP1 protein in vivo in a surrogate wound healing model as well as in in vitro angiogenesis assays. An additional question posed was regarding whether the FSP1 protein itself has a pro-angiogenic role. Although the specific role of FSP1 protein in promoting angiogenesis has been documented by many groups in tumor angiogenesis (Angiogenesis. 2014 Jan;17(1):17-26, The Journal of Biological Chemistry. 2005 May; Vol. 280, No. 21, pp. 20833–20841, Oncogene. 2004 July; volume 23, pages 5487-5495), its role in blood vessel formation in wound repair has not been reported. To assess the role of FSP1 protein itself we attempted to knock down FSP1 in FSP1-fibroblasts but were unsuccessful due to technical difficulties. However, we successfully evaluated the effect of recombinant FSP1 protein (R&D) in vitro on promoting HUVEC proliferation and tube formation (**Please see revised Fig. 7A, B, and C**). As expected, FSP1 protein promoted HUVEC proliferation and tube formation, as previously reported. However, the effects of rFSP1 on HUVEC proliferation and tube formation were significantly reduced as compared to conditioned media from FSP1-expressing cells. These data suggest that the pro-angiogenic effects of the secretome of FSP1-expressing fibroblasts in vitro is due to combined effects of FSP1 protein itself as well as other factors..

4. There are no FSP1+ fibroblasts in normal hearts, but there is a significant increase in their numbers following infarction. Are these cells derived from resident fibroblasts? If so, which signals upregulated FSP1 expression in fibroblasts? The mechanism of activation likely does not involve TGF-beta (considering that this mediator promotes myofibroblast conversion). Limited in vitro work may be sufficient to address this concern.

As suggested by the reviewer, we treated uninjured cardiac fibroblasts with growth factors known to be upregulated following injury and checked for FSP1 expression by real time RTPCR. As expected TGFβ had no effect on FSP1 transcript induction whereas Wnt3a and PDGF modestly induced FSP1 expression. Interestingly, statistically significant increase in FSP1 expression was identified in the presence of FGF2 (**Please see revised Fig. S6 and text on page 11 and 12**). The molecular mechanism regulating FSP1 expression by FG2 would be explored in future studies.

5. The proposed reparative role of FSP1+ fibroblasts in cardiac healing is interesting, but is not documented. Considering the challenges in demonstrating such an effect in vivo, the authors should recognize this limitation and tone down related statements.

Thank you for pointing out this shortcoming. We have revised the manuscript accordingly.

Reviewer #2 (Remarks to the Author):

Saraswati et al

Fibrosis is a significant problem associated with tissue dysfunction. Until recently the fibroblast population has been difficult to study due to a paucity of adequate markers. In this manuscript the authors use two distinct transgenic mouse lines to investigate phenotypic differences in

fibroblasts. In their results the authors suggest that injury induced fibroblasts that express Fibroblast specific protein 1 have different properties than fibroblasts expressing α SMA. A primary distinction is that FSP1 fibroblasts are pro-angiogenic while α SMA expressing cells are not.

Major comments

The authors provide an interesting perspective on fibroblast subpopulations. They provide data suggesting that FSP1 positive cells are characteristically different from the typical myofibroblast α SMA positive cell. One important detail is that the authors need to be sure that there are no contaminating cells in their sorted population. While the authors demonstrate little contamination in figure 2, further validation should be used to demonstrate that this is true for all of the populations of cells with every sorting. Possibilities would be qPCR for CD31, PECAM, CD45, and other immune cell markers through out the experiments.

FACS isolation of FSP1 fibroblasts from cardiac tissue homogenates used CD31 and CD45 antibodies to exclude both endothelial, monocytic and hematopoietic cells, which may also express FSP1, in order to enrich for FSP1 fibroblasts. Sorted FSP1-GFP⁺/CD45⁻/CD31⁻ cells also stained negative for CD45 and CD31 markers (immunofluorescence). In the revised manuscript, we have performed additional experiments to exclude pericytes (NG2/AN2 positive cells) in our cell sorting. FACS sorted FSP1 positive/CD45 and CD31 negative cells were negative for the pericyte marker, NG2, as assessed by flow cytometry as well as by RT-PCR. We used retinal pericytes and MSCs as positive controls for pericytes.

Additionally, the materials and methods state that primary fibroblasts P0-P5 were seeded. For each experiment they should verify the passage of fibroblast used as it in most cases they state freshly isolated so it is unclear when the passaged cells are used. In addition, validation of findings should be shown by additional means as suggested below.

Throughout the paper, we have mostly used freshly sorted cells unless needed to generate enough protein or RNA for some experiments. In the revised manuscript we have specified the passage of the cells utilized for the respective experiments.

In addition there are a few additional points that should be addressed:

The authors provide statistical evaluation of their data, but it is often difficult to discern how n values are counted. For example, the protein array had SD listed, but the authors state that there is an n of 2. Are the stats calculated by n of two in duplicate or are there only two data points being used for each fibroblast population. The authors should clearly state if the n value is a biologic replicate or not. More details on reproducibility would strengthen the manuscript.

We have clarified the “n’s” in the manuscript.

The separation of GFP positive and negative cells is not clear from the flow diagrams. How did the investigators determine what population to identify as positive? Did the investigators include a live dead stain in their flow analysis?

For each flow sorting, gates were established using unstained uninjured fibroblasts, and single stains (GFP for FSP1 and α SMA, CD31-PE, CD45-APC and 7AAD for dead cells). GFP positive cells which were negative for 7AAD, APC, and PE were sorted to obtain FSP1 fibroblast population from FSP1-GFP mice heart. Whereas, GFP positive and 7AAD negative cells were sorted from α SMA-GFP mice.

Also the authors should describe how the images in 2D were obtained using HRP conjugated antibodies.

This was an error in the manuscript. The immunofluorescence was performed using fluoro-chrome conjugated primary or secondary antibodies. The details have been included in the revised manuscript.

The cultures appear surprisingly homogeneous in figure 2D. Are these freshly plated cells from the flow plots after just 4 hours?

The cells were seeded following sorting and were cultured under they reached around 80% confluency prior to immunostaining.

The authors should include information regarding what passage of cells were used for each experiment.

In most experiments, cells were freshly sorted. However, we have clarified the passage of cells throughout the manuscript.

It would be informative for the authors to report % of fibroblasts using the respective transgenic markers in addition to staining for PDGFR α and mEF-SK4 antibody by flow cytometry two markers that have been reported previously for cardiac fibroblasts to determine if these two populations constitute the majority of cardiac fibroblasts.

The authors thank the reviewer for suggesting mEF-SK4 as pan-fibroblast marker. We have included data which analyzed the isolated uninjured and sorted GFP positive FSP1 and α SMA fibroblasts. The results are presented in the revised manuscript (**Please see revised Fig. 2E**).

As the α SMA GFP line has not been extensively described in the heart. The authors should perform some characterization of the GFP expression similar to what was performed for figure 1 and α SMA staining. What percent of α SMA cells are GFP positive under these various conditions? Are GFP expression and α SMA protein synonymous?

In figure 1 the mice used are wildtype C57/Bl6, hence there was no GFP expression. The transgenic α SMA-GFP mice used in the study have been well characterized to demonstrate that cells that express α SMA also express GFP (KALAJZIC, 2008, *Bone*, 43, 501-10).

For all uninjured experiments is based on what is known about α SMA expression, it seems like the authors are comparing VSMC/pericytes to FSP1+ cells rather than uninjured fibroblasts. The authors need to address why the α SMA population in an uninjured heart would be considered a fibroblast.

The authors apologize for any confusion. The only α SMA population that exists in uninjured heart are perivascular cells or vascular smooth muscle cells.

QPCR for α SMA would support the immunofluorescence assay in fig 3 and provide quantitative data. Also the authors should add α SMA fibroblasts to the experiment.

We have added real time RT-PCR analysis of α SMA to revised Fig.4 as well as immunostained α SMA fibroblast in the revised manuscript.

It is odd that the authors did not validate the expression of proteins observed in the protein array. They should consider verifying 1-2 of the proteins that are most differentially expressed in 5A and B.

We confirmed the secretion of VEGF A and Gremlin 1 protein in the secretome (conditioned media) by ELISA. In the revised manuscript we also performed real time RT-PCR for *fgf1*, *Angiopoetin 1*, *Vegf a* and *Vegf b* in addition to *Gremlin1* to demonstrate differential transcript expression in different fibroblast subtypes. Briefly, the pro-angiogenic genes *Fgf1*, *Angiopoetin 1*, *Vegf a* and *Vegf b* were remarkably upregulated in FSP1 fibroblasts in comparison to uninjured and α SMA fibroblasts (**Please see revised Fig. 6D**).

There are also some discrepancies in data for example Fig 4 shows that thrombospondin 2 RNA is upregulated in α SMA fibroblasts while thrombospondin 2 protein is predominantly in the FSP1 protein array. How do the authors explain these contradictions?

The authors apologize for this oversight. Since thrombospondin 2 data is not the crucial part of this paper, we have removed the thrombospondin 2 data from the protein array to avoid confusion.

Minor comments

As it is very germane to this manuscript primary references should be provided on page 3 line 70 rather than a review.

We have provided specific references.

The table does not clearly label the columns from the FSP1+ cells and the α SMA+ cells. We have made those changes.

Typo known page 4 line 92

Thank you for catching that error. We have fixed the typo.

Reviewer #3 (Remarks to the Author):

This is a very elegant and comprehensive study of two distinct fibroblast populations in the heart. The authors make a compelling case for the distinctness of the two populations and their unique transcription profiles. I have a few comments about the data.

1. the focus on pro-angiogenic role of the FSP+ population is unwarranted and only diminishes the paper. First of all, any cell(s) expressing lots of VEGF will be "pro-angiogenic". So the entire data set devoted to the "pro-angiogenic" effect is largely meaningless. You dont need to prove that VEGF induces angiogenesis.

We agree that designating the FSP1 cell population purely based on transcriptome or proteomic data demonstrating elaboration of specific cytokines/chemokines is not very meaningful. However, the basis for our statement, which we feel is both novel and important, is that we demonstrate that FSP1+ fibroblasts induce robust angiogenesis in an in vivo sponge granulation tissue model in a manner significantly distinct from myofibroblasts or fibroblasts isolated from uninjured hearts (Fig.7). We also subsequently demonstrate that conditioned media from FSP-1 cells have distinct positive functional effects on HUVECs in a manner distinct from myofibroblasts and uninjured fibroblasts. It is the strength of these functional in vivo and in vitro data, combined with the proteomic and genomic data that have led us to our statements.

Second, if the authors truly want to make this point, they need to knockout this entire population in the mouse and describe the phenotype. The existence of two distinct sets of fibroblasts is a much bigger story and should the focus.

Although FSP1 and α SMA are proteins that serve to mark these populations, their functional roles are many and not completely understood. Hence, ablating these proteins may not eliminate these cells. We do not yet understand how to eliminate fibroblast subpopulations.

2. It would be interesting to know how uniform these populations are. single cell RNA-seq could provide major biological insights here

This is an excellent suggestion but these experiments are extensive and time-consuming and unfortunately beyond the scope of this study.

3. One wishes for a bit of insight into the origin of these populations, what drives their existence/expression profile differences and their biological roles.

Please see the response to Reviewer 1, query 4.

Reviewers' comments:

Reviewer #1 (Remarks to the Author):

The authors have addressed many of my concerns. Unfortunately, 2 important experiments were not performed. First, functional assays necessary to support the angiogenic phenotype of FSP1+ fibroblasts were not performed. Second, in vitro loss-of-function experiments testing the hypothesis that FSP1 may be required for angiogenic actions (and is not simply a marker) were not successful. However, I feel that the novelty of the concept outweighs any weaknesses in documentation. Thus, I have no further recommendations for the authors.

Reviewer #2 (Remarks to the Author):

The authors have addressed many of the previous concerns. The addition of the angiogenesis results strengthen the manuscript, but the authors should compare and contrast their results with existing literature on FSP1 and angiogenesis.

A key concern is the use of the asma-GFP line to distinguish the fibroblast populations and only at 10 days after LAD ligation. If the reporter does not represent endogenous aSMA protein expression this is a major flaw in interpreting the results. The rebuttal to the question regarding aSMAGFP vs ASMA staining overlap was inadequate. The aSMAGFP has only been "well characterized" in the bone. Additional images in the referenced manuscript were in uninjured adipose, skeletal muscle, liver, and lung. Due diligence should be performed to analyze reporter fidelity by IHC and could even be performed via flow cytometry after LAD ligation. Unless the authors can provide additional references to the validation of this reporter in the heart, they should perform some minimal evaluation.

There are still also some minor issues to address.

Also other than gremlin, the genes identified as angiogenesis genes in figure 5F are not typical angiogenesis genes. These are more classified as inflammatory mediators.

For example the figure legend in Fig S4 says freshly isolated and cultured fibroblasts. It is unclear that the cells could be both. Also are the α SMA and FSP1 positive cells isolated after injury in Figs S4 and S5? In the m&m on page 26 what does the distinction of uninjured fibroblasts mean? Are the GFP populations isolated after injury?

Authors should clarify the statement on page 6 "like many fibroblast markers". Do they mean Thy1 and Sca1?

In the results section they jump from using antibodies in figure 1 to the use of GFP reporter mice in figure 2 with little explanation or reference to the mice and their origination.

The authors state this method for dead cell exclusion "Dead cells were excluded by using 7AAD (Molecular Probes: A13010) or dead cells were washed in culture 4 hours after seeding." Was 7AAD used for all experiments? If the cells are plated, the authors should not refer to them as freshly isolated.

What are the baseline levels of the cell populations by flow cytometry (uninjured)?

The authors cannot conclude that the FSP1 cells that they observe by IHC are fibroblasts and therefore they should temper any discussion relating to time course of cell population. Most studies presented here were using day 10 after LAD ligation and therefore not providing any time course details.

The authors might want to comment on how their FSP1 population compares to recently described distinct cardiac fibroblast populations in manuscripts from Skelly...A. Pinto (cell reports) and J. Molkentin (JCI).

Typos

Figure s4 transcripts

Figure s6 PDGF β should be PDGFBB

Page 43 ANF

Reviewer #3 (Remarks to the Author):

Most issues have been adequately addressed

Authors' response:

Title: Identification of a pro-angiogenic functional role for FSP1 positive fibroblast subtype in wound healing

Please find below author's response to reviewer's comments in blue:

Reviewers' comments:

Reviewer #1 (Remarks to the Author):

The authors have addressed many of my concerns. Unfortunately, 2 important experiments were not performed. First, functional assays necessary to support the angiogenic phenotype of FSP1+ fibroblasts were not performed. Second, in vitro loss-of-function experiments testing the hypothesis that FSP1 may be required for angiogenic actions (and is not simply a marker) were not successful. However, I feel that the novelty of the concept outweighs any weaknesses in documentation. Thus, I have no further recommendations for the authors.

Although, the reviewer is satisfied with the revisions, an additional concern was raised on the functional assays we used to demonstrate angiogenic activity of FSP1 fibroblasts. We summarize the experiments below:

1. Fig 7; Rebuttal figure 1.
 - a. In vitro Endothelial cell assembly assay (Fig 7A and B) and endothelial cell proliferation (Fig 7C).
 - b. In vivo assessment of vascularization in wound repair assay (Fig 7D and E).

We also showed by in vitro proliferation and endothelial cell

assembly assay that FSP1 fibroblast secretome has a much profound impact on endothelial cell proliferation as well as on the ability of endothelial cells to form vascular assembly in comparison to FSP1 protein alone. These observations do not discount the effect of FSP1 protein on angiogenesis but emphasize that there are additional factors in FSP1 secretome that impart the robust pro-angiogenic phenotype of the FSP1 expressing fibroblasts.

Reviewer #2 (Remarks to the Author):

The authors have addressed many of the previous concerns. The addition of the angiogenesis results strengthen the manuscript, but the authors should compare and contrast their results with existing literature on FSP1 and angiogenesis.

The authors thank the reviewer for the suggestion. We have included the discussion of existing FSP1 literature with our data (see page 21).

A key concern is the use of the alpha-sma-GFP line to distinguish the fibroblast populations and only at 10 days after LAD ligation. If the reporter does not represent endogenous aSMA protein expression this is a major flaw in interpreting the results.

We performed a time course following injury of expression of these markers by immunostaining analysis and identified that the expression of both α SMA and FSP1 protein reached its peak around day 8-12 in heart after LAD ligation (presented in Fig.1 and Rebuttal figure 2). Therefore, to augment yield, we picked day 10 as a time point for activated fibroblast isolation following injury.

The rebuttal to the question regarding aSMAGFP vs ASMA staining overlap was inadequate. The aSMAGFP has only been “well characterized” in the bone. Additional images in the referenced manuscript were in uninjured adipose, skeletal muscle, liver, and lung. Due diligence should be performed to analyze reporter fidelity by IHC and could even be performed via flow cytometry after LAD ligation. Unless the authors can provide additional references to the validation of this reporter in the heart, they should perform some minimal evaluation.

We agree with the reviewer that α SMA-GFP reporter line is only well characterized in bone (KALAJZIC, 2008, *Bone*, 43, 501-10). Also, the expression/co-localization of α SMA and GFP was identified in the perivascular region of uninjured adipose, skeletal muscle, liver, and lung not in heart.

In this study we comprehensively analyzed the expression of α SMA protein in the heart. We showed that in the uninjured heart α SMA protein expression is localized to the perivascular region (Figure 1) whereas around 8 day following myocardial injury, α SMA expressing cells also appear in the interstitial space of the injured myocardium.

We used transgenic mouse model which expresses GFP under α SMA promoter to study the function of α SMA expressing fibroblasts. To ensure the purity of GFP expressing α SMA fibroblasts, we seeded the cells on coverslip immediately after GFP sorting and stained them for α SMA, FSP1, COL1a1, Periostin, Vimentin, CD31, and CD45 (Figure 2). Almost all of the GFP positive cells were positive for α SMA, COL1a1, and Periostin and negative for FSP1, CD45, and CD31. These results ensured reporter fidelity by identifying that the GFP expressing cells from α SMA-GFP transgenic mice express α SMA protein as well as markers for fibroblasts and are not endothelial (CD31-) and hematopoietic (CD45-) populations.

Although, we evaluated GFP positive sorted cells for the expression of α SMA, we had not done flow cytometric/IHC evaluation of GFP/ α SMA co-localization. Based on the reviewer's suggestion we performed flow cytometric evaluation of GFP sorted cells from α SMA-GFP mice. We identified close to 99.9% GFP positive cells from α SMA-GFP transgenic mice stain positive for α SMA protein (Rebuttal figure 3).

There are still also some minor issues to address.

Also other than gremlin, the genes identified as angiogenesis genes in figure 5F are not typical angiogenesis genes. These are more classified as inflammatory mediators.

Both IL10/IL10 receptor and IL6/IL6 receptor have been shown by others as angiogenic regulators (PLoS One, 2008 Dace et al. and Oncogene (2003), Wei. Et al.). We have changed the labeling to "angiogenesis modulators" from "angiogenesis" genes.

For example the figure legend in Fig S4 says freshly isolated and cultured fibroblasts. It is unclear that the cells could be both. Also are the α SMA and FSP1 positive cells isolated after injury in Figs S4 and S5? In the m&m on page 26 what does the distinction of uninjured fibroblasts mean? Are the GFP populations isolated after injury?

We have changed the text in Fig S4 (removed freshly isolated and cultured). If seeded on a coverslip/plate, the passage of the cells is mentioned in the paper. α SMA and FSP1 proteins are not expressed in the uninjured heart (Fig 1 A). For all the FSP1 and α SMA cell isolation, MI's were performed in mice and cells were GFP sorted at day 10 following injury. Endothelial cells and hematopoietic cells were excluded from FSP1-GFP sorting. It has been clarified in the Methods section (see page 25).

Authors should clarify the statement on page 6 "like many fibroblast markers". Do they mean Thy1 and Sca1?

We have removed that statement from the paper since it was not necessary for FSP1 marker description.

In the results section they jump from using antibodies in figure 1 to the use of GFP reporter mice in figure 2 with little explanation or reference to the mice and their origination.

The authors apologize for this oversight. We have edited the results section to introduce the transgenic mice prior to their utilization (please see page 6, 7, and 8).

The authors state this method for dead cell exclusion "Dead cells were excluded by using 7AAD (Molecular Probes: A13010) or dead cells were washed in culture 4 hours after seeding." Was 7AAD used for all experiments? If the cells are plated, the authors should not refer to them as freshly isolated.

7AAD was not used for the experiments where cells were plated. Only the live cells attached whereas dead cells were washed away. We have removed the text "freshly isolated" and mentioned the passage of the cells wherever the cells were plated.

What are the baseline levels of the cell populations by flow cytometry (uninjured)?

Unstained uninjured fibroblasts were used to establish the background signal in the GFP channel post compensation. A gate was drawn to capture all cells producing a larger voltage pulse area than the established background in the GFP channel. GFP positive cells which were negative for PE, and APC were sorted to obtain an FSP1 fibroblast population from the hearts of FSP1-GFP mice. For α SMA-GFP mice, live GFP positive cells were selected by gating for GFP positive cells. To clarify this, we have revised Fig 2 where we have added the unstained compensation control from uninjured fibroblasts. Revision in the text is presented on page 7.

The authors cannot conclude that the FSP1 cells that they observe by IHC are fibroblasts and therefore they should temper any discussion relating to time course of cell population. Most

studies presented here were using day 10 after LAD ligation and therefore not providing any time course details.

The time course staining for FSP1 and α SMA protein (Fig 1) as described as “fibroblasts” is changed to “cells”.

The authors might want to comment on how their FSP1 population compares to recently described distinct cardiac fibroblast populations in manuscripts from Skelly...A. Pinto (cell reports) and J. Molkenin (JCI).

The relevant paper has been included in the revised manuscript.

Typos

Figure s4 transcripts

Figure s6 PDGF β should be PDGFBB

Page 43 ANF

We apologize for the typos and have fixed them.

Reviewer #3 (Remarks to the Author):

Most issues have been adequately addressed

Thank you.

Reviewers' comments:

Reviewer #2 (Remarks to the Author):

Because they use the aSMA GFP for cell identification in a majority of the manuscript, a supplemental image demonstrating aSMAGFP after MI in sections would provide important information to the reader regarding this cell population. Maybe day 0, 4, and 10 would be adequate? They show that aSMAGfp are aSMA positive, but what percentage of aSMA positive cells are GFP?

Authors' response:

Title: Identification of a pro-angiogenic functional role for FSP1 positive fibroblast subtype in wound healing

Please find below author's response to reviewer's comments in blue:

Reviewers' comments:

Reviewer #2 (Remarks to the Author):

Because they use the α SMA GFP for cell identification in a majority of the manuscript, a supplemental image demonstrating α SMAGFP after MI in sections would provide important information to the reader regarding this cell population. Maybe day 0, 4, and 10 would be adequate? They show that α SMAGFP are α SMA positive, but what percentage of α SMA positive cells are GFP?

Thank you for allowing us to address one further question. Reviewer 2 is interested in understanding our α SMA-GFP reporter mouse model better. In our last re-submission we provided data that showed **99±0.057%** percentage of GFP+ cells were α SMA+ at day 10 following myocardial infarction (MI). Reviewer 2 would like to also see the converse data where we show the percentage of α SMA+ cells that are also GFP+. To determine that, we evaluated homogenized and digested mouse hearts 10 days after MI by flow as this is the time frame in which we see high expression of α SMA+ cells. We gated first for all α SMA+ cells and determined what percentage of these also demonstrated emission properties consistent with being GFP+. By evaluating three independent experiments in which each experiment contained a pool of three animals from which hearts were isolated and analyzed 10 days post MI showed that a high percentage of α SMA+ cells are also GFP positive (**94±7.1%**, see attached Rebuttal Figure 1 and revised supplementary figure S3 (**Fig S3**). The text is revised in blue in the main manuscript (see page 7 and 8). We also analyzed α SMA and GFP co-expression prior to injury and after 4 days following MI by flow. While negligible number of cells were positive for GFP and α SMA co-staining in uninjured heart, we identified **19.53±0.635%** of α SMA+ cells co-expressing GFP. Similarly, a high percentage of GFP + cells co-express α SMA (**88±0.7%**). Our data suggest that while a very high percentage of GFP+ cells co-express α SMA in all time points tested, the expression of GFP in newly emerging α SMA cells lags temporally. **Importantly, we isolated α SMA+ cells by flow sorting for GFP 10 d after MI, a time frame in which there is a high concordance of all α SMA+ cells expressing GFP and vice versa.** As expected, day 0 had very few cells that were either GFP+ and/or α SMA+. The data are shown graphically below.

Another way to get at this question (which the reviewer suggests) is to perform co-immunostaining of GFP and α SMA following MI. We did not use this approach due to the high background of GFP antibody in our hands during immunostaining, making quantification very difficult and much less reliable. Evaluating a number of sections across animals has inherent bias as well.

Thank you.

Pampee Young, MD, PhD and Sarika Saraswati, PhD

Reviewers' comments:

Reviewer #2 (Remarks to the Author):

Because of the limited time window of their analysis and the fact that the trans gene does not follow gene expression until the 10 day window the authors need to list the limitations of their study and also provided the caveat that the transgene was used to identify the as a cells and that these cells may contain other cell types that could be antiangiogenic. The percent of cells positive for aSMA antibody staining in the data provided to the reviewer also seems to be high with few negative cells. How is this the case if there are two populations of fibroblasts one that is FSP and one that is aSMA. Were these cells preplated before flow cytometry and if so for how long? Looking over the materials and methods, do the authors not exclude CD45+ and CD31+ cells from the aSMAGFP sorts? If not they need to clearly state this in the m&m as well as listing it as another limitation of the study.

Authors' response:

Title: Identification of a pro-angiogenic functional role for FSP1 positive fibroblast subtype in wound healing

Please find below author's response to reviewer's comments in blue:

Reviewer #2 (Remarks to the Author):

Because of the limited time window of their analysis and the fact that the trans gene does not follow gene expression until the 10 day window the authors need to list the limitations of their study and also provided the caveat that the transgene was used to identify the as a cells and that these cells may contain other cell types that could be antiangiogenic.

As suggested by the reviewer we have added the requested caveats and limitations as well as our rationale for time frame of cell isolation into the revised result section on page 8 and the discussion section on page 17. We agree that any transgenic model will have limitations.

There has been a misunderstanding regarding the reviewer's second statement in that we have not made a claim that α SMA cells are anti-angiogenic. Rather, when we directly compare FSP1+ fibroblasts with α SMA+ fibroblasts (which are sorted, highly purified populations), FSP1+ has a significantly greater comparative proangiogenic effect in vitro and in an in vivo assay (mentioned on pages 14 and 16). That is very different from stating that α SMA are anti-angiogenic.

The percent of cells positive for α SMA antibody staining in the data provided to the reviewer also seems to be high with few negative cells. How is this the case if there are two populations of fibroblasts one that is FSP and one that is α SMA. Were these cells preplated before flow cytometry and if so for how long?

We have noted differences in percentage of GFP negative population as an aspect of mouse to mouse variation. For example, a representative figure from a different cell isolation was presented in **Figure 2** as well as **Rebuttal Figure 1** where a substantial number of GFP negative cells were

present while gating GFP/ α SMA as well as GFP/FSP1 populations. To avoid confusion, we have removed the gating scheme from Supplementary Figure 3. For flow analysis, the cells were not plated.

Looking over the materials and methods, do the authors not exclude CD45+ and CD31+ cells from the α SMAGFP sorts? If not they need to clearly state this in the m&m as well as listing it as another limitation of the study.

CD45 and CD31 positive cells are a known contaminant of FSP1+ cells (as myeloid populations express FSP1), hence they were only excluded during sorting for FSP1-GFP cells by FACS.

Since hematopoietic cells do not express α SMA (Yokoto et al. Stem Cells, 2006), they are sorted out during isolation of α SMA-GFP cells. Nevertheless, we assessed by immunostaining whether our flow sorted α SMA+ cells contained contamination by CD31+ and CD45+ cells by immunostaining flow-sorted populations (**Figure 2C**) and determined that they were indeed absent as expected.

In summary, with both FSP1 and α SMA cell isolations and characterization we have taken substantial measures to ensure cellular as well as functional purity. Regardless, the caveats associated with transgenic models and flow sorting cannot be overlooked (i.e. in that our assays such as flow and immunostaining are not 100% sensitive to identify rare contaminant). We have made a statement about these caveats in the result section (see page 8). The sorting details are mentioned in the methods section on page 27.

Reviewers' comments:

Reviewer #2 (Remarks to the Author):

The concern with these studies is that they hinge on the distinction of the fibroblasts identified by the two GFP transgenic lines. The FSP1GFP has been used described before and its caveats are clear and the authors address those very well. The concern is that a majority of the data is based on comparing the Fsp1GFP population at 10 days to the asmaGFP population at 10 days. Not present in the current folder but in the previous rebuttal the authors showed a flow plot from 10 day asmaGFP that had very few GFP negative cells from the isolation. This was a point I raised which they rebutted with the answer that there is mouse to mouse variability. Also careful examination of the GFP flow plots shows little distinction between GFP negative and positive cells and all gating is based on a negative control sample.

Minimally the authors should provide % of GFP positive cells out of live cells (using live dead staining) for the aSMA at baseline and after 10 days injury on 5-6 animals without any preplating. They need to indicate for every experiment if 4 hour preplating occurred.

Simultaneous with the above GFP flow the authors should perform flow to determine the number of CD31 and CD45 positive cells in the aSMAGFP population. The ICC in figure 2C is not adequate as these are obviously plated cells and endo and immune cells are less adherent than fibroblasts. This would alleviate the concerns that the aSMAGFP population has a contaminant skewing the gene expression

An IHC image of a day 10 aSMGFP stained for Fsp1 and a day 10 Fsp1GFP stained for aSMA would also alleviate some of the concerns. They have the stains for these working as shown in figure 1. Day 12 staining even suggests that a majority of the aSMA positive cells are VSMC at this time point.

Additional comments in legend of Figure 1 the authors cannot refer to the fsp1 and asma positive cells as fibroblasts in the images. A point that I raised previously.

Scale bar for figure 4A lower panels should be included (the nuclei appear to be different sizes in top panel vs lower panels)

Reviewer's comments:

The concern with these studies is that they hinge on the distinction of the fibroblasts identified by the two GFP transgenic lines.

The FSP1GFP has been used described before and its caveats are clear and the authors address those very well. The concern is that a majority of the data is based on comparing the Fsp1GFP population at 10 days to the asmaGFP population at 10 days. Not present in the current folder but in the previous rebuttal the authors showed a flow plot from 10 day asmaGFP that had very few GFP negative cells from the isolation. This was a point I raised which they rebutted with the answer that there is mouse to mouse variability. Also careful examination of the GFP flow plots shows little distinction between GFP negative and positive cells and all gating is based on a negative control sample.

1. Minimally the authors should provide % of GFP positive cells out of live cells (using live dead staining) for the aSMA at baseline and after 10 days injury on 5-6 animals without any preplating. They need to indicate for every experiment if 4 hour preplating occurred.
2. Simultaneous with the above GFP flow the authors should perform flow to determine the number of CD31 and CD45 positive cells in the aSMAGFP population.

The ICC in figure 2C is not adequate as these are obviously plated cells and endo and immune cells are less adherent than fibroblasts. This would alleviate the concerns that the aSMAGFP population has a contaminant skewing the gene expression

We discussed these requests with the editor to ensure that we understood what was requested and the rationale for the concerns. The editorial board further supported the reviewer's request and also underscored the importance of providing flow cytometric analysis of fresh, un plated heart homogenates from 5-6 uninjured and injured mice of both transgenic strains, with special emphasis on α SMA-GFP strain, to understand and quantify any contaminating populations. We have provided the requested data by analyzing uninjured animal hearts as well as α SMA-GFP and FSP1-GFP mice hearts 10 days after MI by flow cytometry analysis. These data provide the percentage of GFP positive cells out of live cells at baseline (**uninjured; 0.01% for α SMA-GFP mice and 0.08% for FSP1-GFP mice**) and after 10 days following MI (**injured; 14.18% for α SMA-GFP mice and 12.06% for FSP1-GFP mice**) on mice (**Rebuttal Table 1 and 2**). These results are comparable to what others have reported for fibroblasts in heart (Pinto et al. Circulation Research 2015). Importantly and as requested, the cells were freshly isolated from mouse hearts and no pre-plating was done prior to the analyses.

As reported in the previous versions of the paper, a significant amount of GFP+ endothelial (**8.18% GFP+/CD31+ cells**) and hematopoietic cells (**12.7% GFP+/CD45+ cells**) were identified in the injured FSP1-GFP mice hearts (**Rebuttal figure 2 and Rebuttal Table 2**). These cells were **sorted out** during our FSP1-GFP cell isolation and characterization prior to performing our genomic analyses and in vivo sponge assay on this population. On the other hand, analysis of α SMA-GFP mice showed significantly lower contamination with endothelial (**3.8% GFP+/CD31+ cells**) and hematopoietic (**3.18% GFP+/CD45+ cells**) cells (**Rebuttal figure 1 and Rebuttal Table 1**). The genomic analysis of α SMA-GFP fibroblasts did not exclude this small contaminating population. However, we did not identify any canonical endothelial and/or hematopoietic gene transcripts in our data (Fig 5), underscoring this as a very minor

contaminating population. Unplated cells were also used for in vivo sponge assay but we do not believe the potential 3% contamination of endothelial cells in sorted α SMA-GFP fibroblasts altered the overall phenotype as the proangiogenic phenotype was actually observed in the FSP1-GFP fibroblasts which had no contamination of endothelial cells.

For remainder of the studies in which we compared FSP1 vs. α SMA fibroblasts, the cells were seeded and used between P0-P5. We provide convincing data (Fig 2 and 3) that these plated cells (uninjured, FSP1-GFP and α SMA-GFP fibroblasts) did not contain contaminating

endothelial, hematopoietic or vascular endothelial /pericyte cell population by immunofluorescent analysis.

We also analyzed both populations for contaminating vascular smooth cells or/and pericytes in the isolated cells using AN2 as a marker. The FACS analyses identified negligible contamination of vascular smooth cells/pericytes in GFP+ live cells from α SMA-GFP mice hearts (**0.15% GFP+/AN2+cells**) as well as in GFP+/CD31-/CD45- live cells (**0.54% FSP+/CD31-/CD45-/AN2+ cells**) isolated from FSP1-GFP mice hearts.

We have presented the results with gating scheme as rebuttal figure 1 and 2 and rebuttal table 1 and 2 herein and as the supplementary figures S4 and S5 and Supplementary table 1 and 2 in the revised manuscript. The revised text is presented in blue in the revised manuscript (page 8, 10, 26, and 27).

Rebuttal figure 2.

Rebuttal figure 2. Representative gating scheme of FACS analyses demonstrating %GFP+ live cells expressing CD31 or CD45 markers from FSP1-GFP mice uninjured hearts (**B**) or injured hearts (**A**) 10 days following myocardial infarction (MI). GFP+/CD31-/CD45- cells were then analyzed for the expression of pericyte marker AN2. At least 5 separate isolations were done from individual mouse hearts. The graphical quantification of the data is presented (**C**).

Rebuttal Table 1. Tabular presentation of flow cytometry data at baseline and 10 days following myocardial injury in α SMA-GFP mice hearts.

Freshly isolated cardiac cells	GFP+ from live cells	GFP+/AN2+	GFP+/AN2-	GFP+/CD31+	GFP+/CD45+	GFP+/CD31-/CD45-
αSMA-GFP MI (n=5)	15.20%	0.24%	99.40%	2.98%	3.78%	91.60%
	8.79%	0.03%	99.40%	5.05%	4.35%	89.10%
	21.10%	0.09%	99.60%	2.87%	4.76%	91.20%
	15.30%	0.30%	99.10%	3.10%	2.18%	92.80%
	10.50%	0.11%	99.60%	4.98%	0.84%	92.00%
Average	14.18%	0.15%	99.42%	3.80%	3.18%	91.34%
Std Dev	0.0482	0.0011	0.0020	0.0112	0.0164	0.0138
αSMA-GFP Uninjured heart (n=6)	0.02%	0%	100%	11.10%	0%	88.90%
	0%	0%	0%	0%	0%	0%
	0%	0%	0%	0%	0%	0%
	0%	0%	0%	0%	0%	0%
	9.21E-05	0%	100%	25.00%	0%	75.00%
	0.03%	0%	100%	0%	30.00%	70.00%
Average	0.01%	0.00%	50.00%	6.02%	5.00%	38.98%
Std Dev	0.0001	0.0000	0.5477	0.1031	0.1225	0.4315

Rebuttal Table 1. Flow cytometric evaluation of %GFP live cells present in the α SMA-GFP mice hearts at baseline (uninjured; n=6; bottom) or 10 days following myocardial infarction (MI; n=5; top). Percentage of GFP+ cells expressing markers of hematopoietic cells (CD45), endothelial cells (CD31) and pericytes/vascular smooth muscle cell marker (AN2) were also evaluated in both uninjured and injured hearts.

Rebuttal Table 2. Tabular presentation of flow cytometry data at baseline and 10 days following myocardial injury in FSP1-GFP mice hearts.

Freshly isolated cardiac cells	GFP+ from live cells	GFP+/CD31+	GFP+/CD45+	GFP+/CD45-/CD31-	GFP+/CD31-/CD45-/AN2+
FSP1-GFP MI (n=5)	17.00%	5.04%	8.99%	12.20%	0.82%
	12.60%	8.29%	12.00%	12.40%	0.29%
	10.60%	11.10%	12.80%	19.00%	1.32%
	5.31%	7.72%	18.40%	11.40%	0.15%
	14.80%	8.75%	11.30%	17.50%	0.10%
Average	12.06%	8.18%	12.70%	14.50%	0.54%
Std Dev	0.0447	0.0218	0.0349	0.0348	0.0052
FSP1-GFP Uninjured heart (n=5)	0%	0%	0%	0%	0%
	0.02%	0%	0%	0%	0%
	0%	0%	0%	0%	0%
	0.33%	19.70%	18.00%	14.00%	0%
	0.04%	28.00%	32.00%	0%	0%
Average	0.08%	9.54%	10.00%	2.80%	0.00%
Std Dev	0.0014	0.1339	0.1456	0.0626	0.0000

Rebuttal Table 2. Flow cytometric evaluation of %GFP+ live cells present in the FSP1-GFP mice hearts at baseline (uninjured; n=5; bottom) or 10 days following myocardial infarction (MI; n=5; top). Percentage of GFP+ cells expressing markers of hematopoietic cells (CD45) and endothelial cells (CD31) were also evaluated. Gated GFP+/CD45-/CD31- cells were analyzed for the presence of pericytes/vascular smooth muscle cell marker (AN2) in both uninjured and injured hearts.

3. An IHC image of a day 10 aSMGFP stained for Fsp1 and a day 10 Fsp1GFP stained for aSMA would also alleviate some of the concerns. They have the stains for these working as shown in figure 1. Day 12 staining even suggests that a majority of the aSMA positive cells are VSMC at this time point.

We spoke with Editor, Dr. Gebala, to clarify this request as we were confused. She confirmed that our provision of the flow data to exclude VSMC contamination of the population would suffice.

4. Additional comments in legend of Figure 1 the authors cannot refer to the fsp1 and asma positive cells as fibroblasts in the images. A point that I raised previously. Scale bar for figure 4A lower panels should be included (the nuclei appear to be different sizes in top panel vs lower panels)

We have made the recommended changes in the legend of figure 1. We have also revised Figure 4 by changing the image in the upper panel with the same magnification as the lower panel.

REVIEWERS' COMMENTS:

Reviewer #2 (Remarks to the Author):

The additional data have satisfied my previous concerns.

REVIEWERS' COMMENTS:

Reviewer #2 (Remarks to the Author):

The additional data have satisfied my previous concerns.

We are happy that our work was able to satisfy reviewer's concerns.

We thank the reviewer for careful evaluation of the manuscript.